# Beyond Entity Correlations: Disentangling Event Causal Puzzles in Temporal Knowledge Graphs

**Qian Chen, Jinyu Zhang, Ling Chen**[*]
State Key Laboratory of Blockchain and Data Security
College of Computer Science and Technology, Zhejiang University, Hangzhou, China
`qianchencs@cs.zju.edu.cn, jy_zhang@cs.zju.edu.cn,`
`lingchen@cs.zju.edu.cn`

## Abstract

Existing Temporal Knowledge Graph (TKG) representation learning approaches focus on modeling entity or relation correlations. However, since TKG datasets are constructed from events, which inherently contain heterogeneous causalities, *focusing solely on entity or relation level correlations is inadequate for event prediction in TKGs*. Although a TKG structural causal model can be established as a theoretical framework for event level causality disentangling, practical disentanglement is non-trivial due to the lack of explicit supervision signals. To this end, we propose a **H**eterogeneous **E**vent causality **D**isentangling **R**epresentation learning **A**pproach (**HEDRA**) for TKG reasoning, which is the first work that *focuses on disentangling heterogeneous causalities at the event level* in TKGs. Specifically, a counterfactual detector module is proposed to disentangle non-causality by leveraging event importance and distributional discrepancies of event representations. Moreover, an Instrumental Variable (IV)-guided disentangling module is proposed to disentangle spurious causality by constructing IVs, which can produce robust event representations against spurious causality through multi-view causality subgraphs. Finally, an evolutionary orthogonal module is proposed to separate dynamic causality from static causality for event prediction. Comprehensive experiments on five real-world datasets demonstrate that HEDRA achieves the state-of-the-art performance.

## 1 Introduction

Temporal Knowledge Graphs (TKGs) are dynamic graphs composed of events $(s, r, o, t)$, where $s$ and $o$ denote subject and object entities, $r$ specifies the relation between them, and $t$ indicates the timestamp (Chen & Chen, 2024). TKG representation learning maps temporally evolving entities and relations into a continuous low-dimensional vector space to capture both temporal evolution and structural information in TKGs (Li et al., 2022). The event prediction task then leverages the representations learned from historical events to infer which relations are likely to occur between entities in the future. TKG representation learning underpins downstream applications, e.g., knowledge reasoning and anomaly detection (Saxena et al., 2021). The variety of correlations among entities and the complexity of temporal patterns make effective TKG representation learning challenging.

Existing TKG representation learning approaches focus on modeling correlations among entities or relations. Some approaches construct entity graphs, where entities serve as nodes and relations between them serve as edges, and learn representations based on graph reachability (Li et al., 2022; Bai et al., 2023; Chen et al., 2024b; Zhang et al., 2024). Other approaches introduce derived structures, e.g., entity groups, hypergraphs, and evolutionary clusters, to capture high-order correlations among entities or relations that are not directly connected (Zhang et al., 2022; Tang & Chen, 2024; Tang et al., 2024; Chen & Chen, 2024). However, since TKG datasets are constructed from events, which inherently contain heterogeneous causalities, *focusing solely on entity or relation level correlations is inadequate for event prediction in TKGs*.

---

[*]Corresponding author.

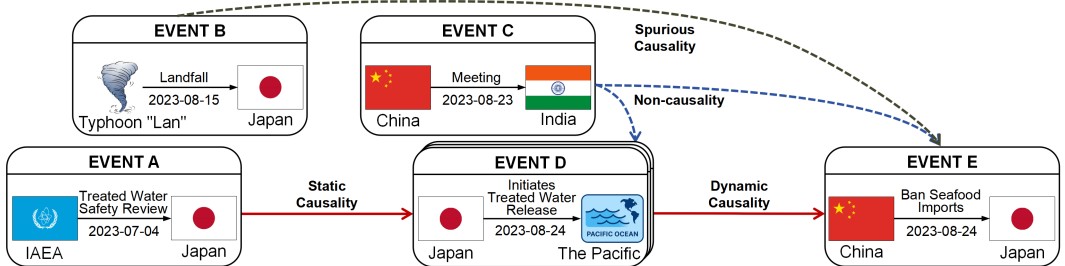

Figure 1: An example of heterogeneous causalities at the event level. IAEA denotes the International Atomic Energy Agency.

In fact, static causality and dynamic causality are ubiquitous in dynamic graphs. Here, static causality refers to time-invariant causal dependencies, whereas dynamic causality captures time-dependent causal dependencies that evolve across timestamps (Zhao & Zhang, 2024). In addition, TKGs also include non-causality that does not benefit event prediction and spurious causality that impedes the model's acquisition of causally relevant discriminative information for event prediction. Figure 1 illustrates heterogeneous causalities at the event level, i.e., static causality, dynamic causality, non-causality, and spurious causality. First, the IAEA comprehensive safety review (Event A) provided the institutional framework that underpinned Japan's decision to begin the treated water release (Event D), exemplifying static causality. Second, when Japan initiated the discharge, China announced an immediate ban on Japanese seafood imports on the same day (Event E), illustrating dynamic causality. Third, the BRIC, i.e., an international organization comprising Brazil, Russia, India, and China, summit held in Johannesburg (Event C) is non-causality with respect to these events. Fourth, Typhoon "Lan" made landfall in Japan and disrupted transport (Event B). Overemphasis on Event B could mislead the model to attribute export changes to Typhoon effects rather than the policy driven seafood ban, demonstrating spurious causality.

Although a structural causal model tailored to event level causalities in TKGs can be posited, it is non-trivial to disentangle static causality, dynamic causality, non-causality, and spurious causality at the event level in TKGs. The challenge lies in identifying and estimating them from observational data, since *existing TKGs lack explicit supervision signals to distinguish these causalities*.

To address the aforementioned challenges, we propose a **H**eterogeneous **E**vent causality **D**isentangling **R**epresentation learning **A**pproach (**HEDRA**) for temporal knowledge graph reasoning. *To the best of our knowledge, HEDRA is the first work that focuses on disentangling heterogeneous causalities at the event level in TKGs*, which constructs event representations from quadruples and progressively disentangles non-causality, spurious causality, static causality, and dynamic causality among TKG events. Our contributions are summarized as follows:

- We propose a TKG structural causal model to formally define non-causality, spurious causality, static causality, and dynamic causality, which establishes a theoretical framework for event level causality disentangling in TKGs.

- We propose a counterfactual detector module to disentangle non-causality in TKGs by leveraging event importance and distributional discrepancies of event representations, which includes a contrastive loss to encourage event pairs with low non-causality weights to be closer and those with high non-causality weights to be more separated.

- We propose an Instrumental Variable (IV)-guided disentangling module to disentangle spurious causality in TKGs by constructing IVs, which can produce robust event representations against spurious causality through multi-view causality subgraphs. In addition, we propose an evolutionary orthogonal module to separate dynamic causality from static causality for downstream event prediction.

- Experiments on five real-world datasets demonstrate that HEDRA achieves the state-of-the-art performance. HEDRA outperforms the runner-up by an average of 5.70%, 7.51%, 7.21%, and 2.30% in MRR, Hits@1, Hits@3, and Hits@10, respectively.

## 2 RELATED WORK

**TKG Representation Learning.** TKG representation learning approaches modeling pairwise correlations typically combine Graph Convolutional Networks (GCNs) and Recurrent Neural Networks to capture structural information and temporal evolution, respectively (Li et al., 2021b; Sun et al., 2021; Li et al., 2022; Bai et al., 2023; Chen et al., 2024b). In addition to pairwise correlations, high-order correlations among three or more entities or relations are modeled through derived structures, i.e., communities, hypergraphs, and evolutionary clusters (Zhang et al., 2022; Tang & Chen, 2024; Tang et al., 2024; Chen & Chen, 2024). However, TKG datasets, e.g., ICEWS, centered on international political events, inherently involve complex causal dependencies. Focusing solely on entity or relation level correlations is therefore inadequate for event prediction in TKGs. To address this, we propose HEDRA, the first framework to disentangle heterogeneous causalities at the event level in TKGs.

**Graph Causality Learning.** Static graph causal learning approaches primarily focus on modeling spatial causality within static graphs. These approaches generally aim to reveal causality by generating interpretable subgraphs (Luo et al., 2020; Yuan et al., 2020; Ying et al., 2019; Fan et al., 2022; Gao et al., 2023). Dynamic graphs, prevalent in real-world scenarios, have motivated research on dynamic graph causal learning, which typically explores both spatial causality and temporal causality (Zhao & Zhang, 2024; Chen et al., 2024a). Despite progress in dynamic graph causal learning, most approaches mainly model static and dynamic causalities while overlooking spurious causality, which impedes the acquisition of causally relevant information for event prediction. Moreover, there is no theoretical framework to disentangle heterogeneous causalities at the event level in TKGs. To fill this gap, we propose HEDRA, which constructs event representations from quadruples and progressively disentangles non-causality, spurious causality, static causality, and dynamic causality among events in TKGs. The comprehensive related works can be found in Appendix A.

## 3 PRELIMINARIES

### 3.1 DEFINITIONS

**Definition (TKG).** A TKG is defined as a sequence of timestamped events, each represented as $\mathcal{G} = \{(s, r, o, t) | s \in \mathcal{E}, \ r \in \mathcal{R}, \ o \in \mathcal{E}, \ t \in \mathcal{T}\}$, where $\mathcal{E}$, $\mathcal{R}$, and $\mathcal{T}$ denote the sets of entities, relations, and timestamps, respectively. $\mathcal{G}^t$ denotes the set of events at timestamp $t$.

**Task (Event Prediction).** The event prediction task in TKGs aims to estimate the probability distribution over candidate relations between a subject entity $s$ and an object entity $o$, conditioned on the historical event sequence $\mathcal{G}^{1:T-1}$. Formally, this task can be expressed as $p(\hat{\boldsymbol{r}}|s, o, \mathcal{G}^{1:T-1})$, where $T$ denotes the total number of historical timestamps.

### 3.2 EVENT LEVEL TKG STRUCTURAL CAUSAL MODEL

The event level TKG structural causal model (SCM) is shown in Figure 2, which comprises nine variables: TKG $\mathcal{G}$, historical events $\mathcal{H}$, non-causality $\mathcal{N}$, causality $\mathcal{C}$, spurious causality $\mathcal{P}$, dynamic causality $\mathcal{D}$, static causality $\mathcal{S}$, representation $\mathcal{R}$, and prediction $\mathcal{Y}$. The directed edges denote cause-effect. The explanations of the TKG SCM are as follows:

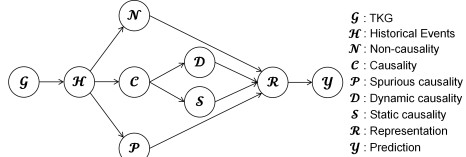

Figure 2: Event level TKG SCM.

- $\mathcal{G} \rightarrow \mathcal{H} \rightarrow \mathcal{C}$: Historical events $\mathcal{H}$ are derived from the TKG $\mathcal{G}$, within which the genuine causal regularities are distilled as causality $\mathcal{C}$.

- $\mathcal{P} \leftarrow \mathcal{H} \rightarrow \mathcal{N}$: Historical events $\mathcal{H}$ give rise to both non-causality $\mathcal{N}$ and spurious causality $\mathcal{P}$. Specifically, $\mathcal{N}$ denotes correlations irrelevant to event prediction, while $\mathcal{P}$ denotes correlations that obstruct the model from capturing causally relevant discriminative information.

- $\mathcal{D} \leftarrow \mathcal{C} \rightarrow \mathcal{S}$: Causality $\mathcal{C}$ can be disentangled into dynamic causality $\mathcal{D}$ and static causality $\mathcal{S}$, both of which jointly drive the learning of high-quality entity and relation representations $\mathcal{R}$.

- $\mathcal{N} \rightarrow \mathcal{R} \leftarrow \mathcal{P}$: Non-causality $\mathcal{N}$ and spurious causality $\mathcal{P}$ interfere with the representation learning process, introducing noise and consequently diminishing the quality of representation $\mathcal{R}$.

- $\mathcal{R} \to \mathcal{Y}$: The objective of HEDRA is to leverage the learned representations $\mathcal{R}$ for event prediction, where the target $\mathcal{Y}$ is ideally determined by dynamic causality $\mathcal{D}$ and static causality $\mathcal{S}$ through $\mathcal{R}$, with the effects of non-causality $\mathcal{N}$ and spurious causality $\mathcal{P}$ suppressed.

From the TKG SCM, two backdoor paths exist between $\mathcal{C}$ and $\mathcal{Y}$, i.e., $\mathcal{C} \leftarrow \mathcal{H} \to \mathcal{N} \to \mathcal{R} \to \mathcal{Y}$ and $\mathcal{C} \leftarrow \mathcal{H} \to \mathcal{P} \to \mathcal{R} \to \mathcal{Y}$, where $\mathcal{N}$ and $\mathcal{P}$ act as confounders that bias the estimation of the causal effect $\mathcal{C} \to \mathcal{Y}$ and thereby interfere with event prediction. Moreover, when estimating the effect of $\mathcal{D}$ on $\mathcal{Y}$, the path $\mathcal{D} \leftarrow \mathcal{C} \to \mathcal{S} \to \mathcal{R} \to \mathcal{Y}$ makes $\mathcal{S}$ a confounder; symmetrically, when estimating the effect of $\mathcal{S}$ on $\mathcal{Y}$, the path $\mathcal{S} \leftarrow \mathcal{C} \to \mathcal{D} \to \mathcal{R} \to \mathcal{Y}$ makes $\mathcal{D}$ a confounder.

**Backdoor Adjustment.** HEDRA aims to progressively disentangle non-causality, spurious causality, static causality, and dynamic causality, ultimately leveraging static and dynamic causalities for event prediction. To this end, it is essential to block backdoor paths so that the model focuses on the causal effect of $\mathcal{C}$. Within this framework, do-calculus (Pearl et al., 2000) provides a principled tool to eliminate the influence of confounders. Specifically, when estimating the causal effect of $\mathcal{C}$ on $\mathcal{Y}$, adjustments for $\mathcal{N}$ and $\mathcal{P}$ are required, while estimation of the effect of $\mathcal{D}$ on $\mathcal{Y}$ requires adjusting for $\mathcal{S}$. Formally, the backdoor adjustment is expressed as (see Appendix B for derivation):

$$P(\mathcal{Y}|\operatorname{do}(\mathcal{D})) = \sum P(\mathcal{Y}|\operatorname{do}(\mathcal{D}), \mathcal{S}) \, P(\mathcal{S}|\operatorname{do}(\mathcal{D})) = \sum P(\mathcal{S}) \sum P(\mathcal{T}) \sum P(\mathcal{P}) \sum P(\mathcal{Y}|\mathcal{G}). \quad (1)$$

However, implementing such backdoor adjustments in TKGs poses significant challenges, since existing TKGs lack explicit supervision signals to distinguish non-causality, spurious causality, static causality, and dynamic causality.

**Our Solution.** To address the above challenge, we propose HEDRA, a framework designed to progressively disentangle heterogeneous event causalities in TKGs. At each timestamp, entity and relation representations are updated through a relation-aware GCN, followed by event representation construction. The counterfactual detector module disentangles non-causality by leveraging event importance and distributional discrepancies, guided by a contrastive loss. The Instrumental Variable (IV)-guided disentangling module introduces IVs to disentangle spurious causality, with a robustness loss based on multi-view causality subgraphs. The evolutionary orthogonal module further disentangles dynamic and static causalities under orthogonality constraints, while an evolutionary loss preserves the temporal dependence of dynamic components and the temporal independence of static components. Finally, by modeling dynamic and static causalities across timestamps, an event graph is constructed and processed with event GCNs to refine entity and relation representations for event prediction. The framework of HEDRA is illustrated in Figure 3.

## 4 Methodology

### 4.1 Event Representation Construction Module

**Relational Message Passing.** Structural dependencies among entities and relations at timestamp $t$ are modeled by a relation-aware GCN as: $h_{\mathrm{o}}^{t,l+1} = \mathrm{RReLU}\left(\frac{1}{d_{\mathrm{o}}} \sum_{(s,r) \in \mathcal{N}_{\mathrm{o}}^{t}} W_1\left(h_{\mathrm{s}}^{t,l} + h_{\mathrm{r}}^{t,l}\right) + W_2 h_{\mathrm{o}}^{t,l}\right)$, where $\mathcal{N}_{\mathrm{o}}^{t} = \{(s,r)|(s,r,o) \in \mathcal{G}^{t}\}$ denotes the set of subject–relation pairs forming incoming edges to object at timestamp $t$. $h_{\mathrm{s}}^{t,l}$, $h_{\mathrm{o}}^{t,l}$, and $h_{\mathrm{r}}^{t,l}$ represent the layer-$l$ representations of subject, object, and relation at timestamp $t$, respectively. $W_1$ and $W_2$ are learnable parameters for neighbor aggregation and the self-loop, respectively. $d_{\mathrm{o}}$ is the in-degree of object. Entity and relation representations are randomly initialized. For brevity, the layer index $l$ is omitted in subsequent sections.

**Relation Update.** The relation representation at timestamp $t$ is influenced jointed by the $r$-related entity representations at the same timestamp and its historical representation. Formally, let $h_{\mathcal{V}_{\mathrm{ent}(\mathrm{r})}}^{t}$ denote the $r$-related entity representations at timestamp $t$. The update of relation representation is formulated as $h_{\mathrm{r}}^{t} = \mathrm{pool}\left([\, h_{\mathcal{V}_{\mathrm{ent}(\mathrm{r})}}^{t}; \, h_{\mathrm{r}}^{t-1} \,]\right)$, where $[;\,]$ denotes concatenation. $\mathrm{pool}$ is mean pooling.

**Event Representation Construction.** To capture the semantic interactions among subjects, relations, and objects at each timestamp, event representations are constructed by jointly encoding the three components of a TKG fact. Formally, for the quadruple $(s, r, o, t)$ with subject representation $h_{\mathrm{s}}^{t}$, relation representation $h_{\mathrm{r}}^{t}$, and object representation $h_{\mathrm{o}}^{t}$, the event representation is formulated as $h_{\mathrm{event}}^{t} = f_{\mathrm{MLP}}\left([\, h_{\mathrm{s}}^{t}; \, h_{\mathrm{r}}^{t}; \, h_{\mathrm{o}}^{t} \,]\right)$, where $f_{\mathrm{MLP}}$ denotes a multi-layer perceptron.

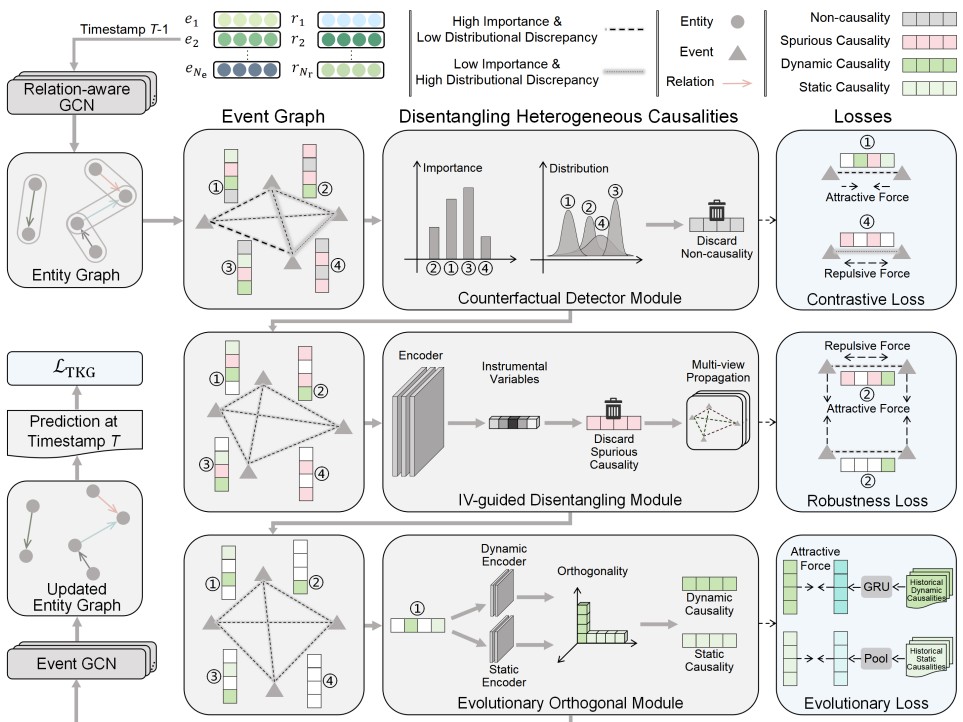

Figure 3: The framework of HEDRA, illustrated with timestamp $T-1$ as an example.

## 4.2 COUNTERFACTUAL DETECTOR MODULE

**Candidate Graph.** To avoid the quadratic complexity of fully connecting all event pairs, a candidate graph is constructed through $k$-nearest-neighbors (Cover & Hart, 1967) in the representation space. Let $\{\boldsymbol{h}_{\text{event},i}^t\}_{i=1}^E$ denote the set of event representations at timestamp $t$, where $E$ is the number of events. The binary adjacency matrix is $\boldsymbol{C} \in \{0,1\}$, with $C_{ij} = 1$ indicating a candidate edge $i \to j$.

**Event Importance.** Higher edge importance implies increased probability of causal dependency between events. For each candidate edge $i \to j$, the event importance is computed as:

$$e_{ij} = \text{LeakyReLU}\Big(\boldsymbol{a}^\top\big[\boldsymbol{W}_3\boldsymbol{h}_{\text{event},i}^t;\ \boldsymbol{W}_4\boldsymbol{h}_{\text{event},j}^t\big]\Big), \quad A_{ij} = \frac{\exp(e_{ij})}{\sum_{k\in\mathcal{N}^{\text{in}}(j)}\exp(e_{kj})}, \tag{2}$$

where $\boldsymbol{W}_3, \boldsymbol{W}_4$, and $\boldsymbol{a}$ are learnable parameters. $\mathcal{N}^{\text{in}}(j) = \{k\,|\,C_{kj} = 1\}$ means the in-degree of event $j$. $A_{ij} \in [0,1]$ is the importance weight on edge $i \to j$.

**Distributional Discrepancy.** Larger distributional discrepancy between event representations implies decreased probability of causal dependency. Each event representation is mapped to parameters of a diagonal Gaussian posterior through multi-layer perceptrons $f_\mu$ and $f_\sigma$ as:

$$\boldsymbol{\mu}_i = f_\mu\left(\boldsymbol{h}_{\text{event},i}^t\right), \qquad \boldsymbol{\sigma}_i = \text{softplus}\left(f_\sigma(\boldsymbol{h}_{\text{event},i}^t)\right), \tag{3}$$

where $\boldsymbol{\mu}_i$ and $\boldsymbol{\sigma}_i$ are the mean and standard deviation vectors of the Gaussian $q_i = \mathcal{N}(\boldsymbol{\mu}_i, \text{diag}(\boldsymbol{\sigma}_i^2))$, respectively. For a candidate edge $i \to j$, the distributional discrepancy is the Kullback–Leibler divergence (Kullback & Leibler, 1951) between $q_i$ and $q_j$ as:

$$D_{ij} = \text{KL}\left(q_i \,\|\, q_j\right) = \frac{1}{2}\sum_{d=1}^D\left[\log\frac{\sigma_{j,d}^2}{\sigma_{i,d}^2} + \frac{\sigma_{i,d}^2 + (\mu_{i,d} - \mu_{j,d})^2}{\sigma_{j,d}^2} - 1\right], \tag{4}$$

where $D$ is the dimension of representations.

**Non-causality Mask.** Event importance and distributional discrepancy are fused on the candidate graph to produce a soft non-causality mask as:

$$\boldsymbol{S} = \left(\alpha_{\text{attn}} \cdot \text{logit}(\boldsymbol{A} + \varepsilon) - \beta_{\text{KL}} \cdot \boldsymbol{D}\right) \odot \boldsymbol{C}, \qquad \boldsymbol{M}^{\text{NC}} = \boldsymbol{1} - \sigma(\boldsymbol{S}), \tag{5}$$

where $\alpha_{\text{attn}}$ and $\beta_{\text{KL}}$ are fixed loss-weight coefficients satisfying $\alpha_{\text{attn}} + \beta_{\text{KL}} = 1$ and are set to 0.5 in all experiments to give equal weight to event importance and distributional discrepancy. $\varepsilon$ is a small constant for numerical stability. $\sigma$ is the sigmoid function and logit is its inverse. $\odot$ denotes the Hadamard product. $\mathbf{1}$ is the all ones matrix. The larger $A_{ij}$ decreases $M_{ij}^{\text{NC}}$ and indicates higher causal dependency likelihood, whereas the larger $D_{ij}$ increases $M_{ij}^{\text{NC}}$ and indicates higher non-causality likelihood.

**Contrastive Loss.** A contrastive loss is proposed to encourage event pairs with low non-causality weights to be closer and those with high non-causality weights to be more separated. Let $s_{ij} = \cos\left(\boldsymbol{h}_{\text{event},i}^t, \boldsymbol{h}_{\text{event},j}^t\right)$ denote cosine similarity, let $\tau > 0$ be the temperature, and let $\mathcal{P} = \{(i, j) \mid i \neq j, \ C_{ij} = 1\}$ be the set of candidate pairs. The contrastive loss is formulated as:

$$\mathcal{L}_{\text{con}} = \frac{1}{|\mathcal{P}|} \sum_{(i,j) \in \mathcal{P}} \left[ (1 - M_{ij}^{\text{NC}})\left(-\log \sigma(s_{ij}/\tau)\right) + M_{ij}^{\text{NC}}\left(-\log(1 - \sigma(s_{ij}/\tau))\right) \right]. \tag{6}$$

### 4.3 IV-GUIDED DISENTANGLING MODULE

**Instrumental Variable Score.** After non-causality has been disentangled, edges between event pairs may still mix genuine and spurious causalities. In causal learning, the Instrumental Variable (IV) approach is commonly employed to disentangle them (Gao et al., 2023). Accordingly, an IV encoder $f_{\text{IV}}$, implemented as a multi-layer perceptron, is adopted to produce the IV score for each edge $i \rightarrow j$ as:

$$\Pi_{ij} = f_{\text{IV}}\left(\boldsymbol{h}_{\text{event},i}^t, \ \boldsymbol{h}_{\text{event},j}^t, \ \text{logit}(A_{ij} + \varepsilon), \ -D_{ij}\right). \tag{7}$$

Herein, only standard TKG quadruples are observed, and no causality labels are available. The IV encoder produces scores that are used solely inside the IV-guided disentangling module (IVDM) to separate genuine from spurious causalities. This design enforces a neural analogue of the exclusion restriction, since $\Pi_{ij}$ does not directly enter the final scoring function. Conditioned on the observed history graph and current event representations, $\Pi_{ij}$ is assumed to be approximately independent of the unobserved noise in the genuine-versus-spurious split, playing the role of the standard IV independence assumption.

**Spurious Causality Mask.** Let $\boldsymbol{M}^{\text{C}} = \mathbf{1} - \boldsymbol{M}^{\text{NC}}$ denote the complement of the non-causality mask. Based on the IV score, the gated matrix $\widetilde{\boldsymbol{\Pi}} = \boldsymbol{M}^{\text{C}} \odot \boldsymbol{\Pi}$ is employed to disentangle genuine and spurious causalities as:

$$\theta_\alpha = \text{Quantile}_\alpha\left(\{\widetilde{\Pi}_{ij} : M_{ij}^{\text{C}} > 0\}\right), \qquad \boldsymbol{M}^{\text{P}} = \mathbb{I}\{\widetilde{\boldsymbol{\Pi}} \geq \theta_\alpha \mathbf{1}\}, \qquad \overline{\boldsymbol{M}}^{\text{P}} = \mathbf{1} - \boldsymbol{M}^{\text{P}}, \tag{8}$$

where $\alpha \in (0, 1)$ specifies the fraction of top-scoring edges. $\theta_\alpha$ is the $\alpha$-quantile of IV scores. $\boldsymbol{M}^{\text{P}}$ selects the top-$\alpha$ edges as genuine causality while $\overline{\boldsymbol{M}}^{\text{P}}$ retains the remainder as spurious causality.

**Multi-view Propagation.** Although the spurious causality mask partitions the edges into genuine and spurious causalities, the IV scores may be imperfect in practice. To enhance robustness, three subgraphs are constructed: the genuine-view subgraph $G_{\text{gen}}$ with $\boldsymbol{M}^{\text{P}}$, the spurious-view subgraph $G_{\text{spur}}$ with $\overline{\boldsymbol{M}}^{\text{P}}$, and the all-view subgraph $G_{\text{all}}$ combining both. Heterogeneous convolution is applied on each subgraph to derive complementary event representations, which is formulated as:

$$\boldsymbol{H}_{\text{gen}}^t = \text{HConv}\left(G_{\text{gen}}, \boldsymbol{H}_{\text{event}}^t\right), \quad \boldsymbol{H}_{\text{spur}}^t = \text{HConv}\left(G_{\text{spur}}, \boldsymbol{H}_{\text{event}}^t\right), \quad \boldsymbol{H}_{\text{all}}^t = \text{HConv}\left(G_{\text{all}}, \boldsymbol{H}_{\text{event}}^t\right), \tag{9}$$

where $\text{HConv}(\cdot)$ denotes heterogeneous event convolution.

**Robustness Loss.** To enhance robustness under imperfect IV estimation, a robustness loss is designed based on different subgraph propagation views, consisting of alignment and separation terms. The alignment term draws the all-view representations toward the genuine-view representations, while the separation term pushes the spurious-view representations away from the genuine-view representations. This process is formulated as:

$$\mathcal{L}_{\text{rob}} = \lambda_{\text{align}} \underbrace{\frac{1}{E} \sum_{i=1}^{E} \left[-\log \sigma\left(s(\boldsymbol{h}_{\text{all},i}^t, \boldsymbol{h}_{\text{gen},i}^t)/\tau\right)\right]}_{\mathcal{L}_{\text{align}}} + \lambda_{\text{sep}} \underbrace{\frac{1}{E} \sum_{i=1}^{E} \left[-\log\left(1 - \sigma\left(s(\boldsymbol{h}_{\text{spur},i}^t, \boldsymbol{h}_{\text{gen},i}^t)/\tau\right)\right)\right]}_{\mathcal{L}_{\text{sep}}},$$
$$\tag{10}$$

where $\lambda_{\text{align}}$ and $\lambda_{\text{sep}}$ are fixed trade-off coefficients with $\lambda_{\text{align}} + \lambda_{\text{sep}} = 1$ and are set to 0.5 in all experiments to give equal weight to the alignment and separation terms.

## 4.4 Evolutionary Orthogonal Module

**Static and Dynamic Causalities Masks.** To disentangle static and dynamic causalities, each event representation is projected into a static component and a raw dynamic component by two multi-layer perceptron encoders $f_{\text{stat}}$ and $f_{\text{dyn}}$ as:

$$h_{\text{event},i}^{S,t} = f_{\text{stat}}(h_{\text{gen},i}^t), \qquad h_{\text{event},i}^{\text{raw},D,t} = f_{\text{dyn}}(h_{\text{gen},i}^t). \tag{11}$$

Then, the dynamic component is obtained by orthogonalizing the raw dynamic component with respect to the static component as:

$$h_{\text{event},i}^{D,t} = h_{\text{event},i}^{\text{raw},D,t} - \frac{\langle h_{\text{event},i}^{\text{raw},D,t}, h_{\text{event},i}^{S,t} \rangle}{\|h_{\text{event},i}^{S,t}\|_2^2 + \varepsilon} h_{\text{event},i}^{S,t}. \tag{12}$$

A classifier $f_{\text{MLP}}$ is employed to distinguish dynamic from static causality, as follows:

$$p_{ij}^D = \sigma\Big(f_{\text{MLP}}\big(\,[\,|h_{\text{event},i}^{D,t} - h_{\text{event},j}^{D,t}|;\ |h_{\text{event},i}^{S,t} - h_{\text{event},j}^{S,t}|]\big)\Big). \tag{13}$$

Herein, the dynamic causality mask is defined as $M^D = \mathbb{I}\{p_{ij}^D > 0.5\} \odot M^{\text{P}}$, and the static causality mask as $M^S = \mathbf{1} - M^D$.

**Evolutionary Loss.** An evolutionary loss is designed to preserve the temporal dependence of dynamic components and the temporal independence of static components as:

$$\mathcal{L}_{\text{evo}} = \lambda_{\text{dyn}} \underbrace{\frac{1}{|\mathcal{G}|} \sum_{g \in \mathcal{G}} \left\| h_{\text{event},g}^{D,t} - f_{\text{GRU}}(\tilde{h}_{\text{event},g}^{D,1:t-1}) \right\|_2^2}_{\mathcal{L}_{\text{dyn}}} + \lambda_{\text{stat}} \underbrace{\frac{1}{|\mathcal{G}|} \sum_{g \in \mathcal{G}} \left\| h_{\text{event},g}^{S,t} - \bar{h}_{\text{event},g}^{S,1:t-1} \right\|_2^2}_{\mathcal{L}_{\text{stat}}}, \tag{14}$$

where $\lambda_{\text{dyn}}$ and $\lambda_{\text{stat}}$ are fixed trade-off coefficients satisfying $\lambda_{\text{dyn}} + \lambda_{\text{stat}} = 1$ and are set to 0.5 in all experiments to give equal weight to the dynamic and static components. $\mathcal{G}$ is the set of $(s, r, o)$ groups. $\bar{h}_{\text{event},g}^{S,1:t-1}$ denotes the group-level mean of the static components, and $\tilde{h}_{\text{event},g}^{D,1:t-1}$ is the historical memory of the dynamic components.

**Static and Dynamic Causalities Modeling.** Static and dynamic causalities are essential for event prediction in TKGs. To exploit them explicitly, message passing is conducted in the static-view subgraph $G_{\text{stat}}$ and the dynamic-view subgraph $G_{\text{dyn}}$. Heterogeneous convolution is applied to the genuine-view representations $H_{\text{gen}}^t$, and the resulting static-view and dynamic-view representations are fused and normalized to update event representations as:

$$H_{\text{evo}}^t = \text{Norm}\Big(W_5\big[\text{HConv}(G_{\text{dyn}}, H_{\text{gen}}^t); \text{HConv}(G_{\text{stat}}, H_{\text{gen}}^t)\big]\Big). \tag{15}$$

Updated entity and relation representations $\overline{H}_{\text{e}}$ and $\overline{H}_{\text{r}}$ are obtained via inverse event construction.

## 4.5 Event Prediction

ConvTransE (Shang et al., 2019) is employed as the decoder to estimate the relation probability distribution for a given entity pair as $p(\hat{r}|s, o, \mathcal{G}^{1:T-1}) = \sigma\big(\overline{H}_{\text{r}} \text{ConvTransE}(\bar{e}_{\text{s}}, \bar{e}_{\text{o}})\big)$, where $\hat{r}$ denotes the predicted probability vector over relations and $\overline{H}_{\text{r}}$ is the relation representation matrix. The training objective for event prediction is to minimize the cross-entropy loss:

$$\mathcal{L}_{\text{TKG}} = -\frac{1}{N_{\text{S}}} \sum_{i=1}^{N_{\text{S}}} \sum_{j=1}^{N_{\text{r}}} \big(y_{i,j} \log p_{i,j} + (1 - y_{i,j}) \log(1 - p_{i,j})\big), \tag{16}$$

where $N_{\text{S}}$ and $N_{\text{r}}$ denote the number of training samples and relations, respectively. $y_{i,j}$ is the ground-truth label for relation $j$ in sample $i$ (1 if the event occurs, 0 otherwise), and $p_{i,j}$ is the predicted probability. The overall training objective of HEDRA integrates multiple components:

$$\mathcal{L} = (1 - \lambda_{\text{con}} - \lambda_{\text{rob}} - \lambda_{\text{evo}})\mathcal{L}_{\text{TKG}} + \lambda_{\text{con}}\mathcal{L}_{\text{con}} + \lambda_{\text{rob}}\mathcal{L}_{\text{rob}} + \lambda_{\text{evo}}\mathcal{L}_{\text{evo}}, \tag{17}$$

where $\lambda_{\text{con}}$, $\lambda_{\text{rob}}$, and $\lambda_{\text{evo}}$ are scalar coefficients for contrastive, robustness, and evolutionary losses, respectively. All hyperparameters are constrained within $[0, 1]$.

Table 1: The performance of HEDRA and the compared approaches on ICEWS14 and ICEWS18. An asterisk ("*") indicates that HEDRA significantly outperforms the compared approaches based on pairwise t-tests at a 95% confidence level. The best performance is highlighted in **bold**, while the runner-up is underlined.

| Approach | ICEWS14 | | | | ICEWS18 | | | |
|---|---|---|---|---|---|---|---|---|
| | MRR | Hits@1 | Hits@3 | Hits@10 | MRR | Hits@1 | Hits@3 | Hits@10 |
| TTransE (WWW 2018) | 22.36* | 13.41* | 24.40* | 39.64* | 16.39* | 8.43* | 17.21* | 31.72* |
| TA-TransE (EMNLP 2018) | 20.94* | 13.77* | 24.11* | 36.06* | 21.64* | 14.11* | 24.89* | 37.64* |
| RE-NET (EMNLP 2020) | 38.53* | 22.53* | 44.47* | 74.05* | 39.63* | 23.55* | 45.70* | **75.66** |
| Glean (KDD 2020) | 36.07* | 22.17* | 39.52* | 64.75* | 35.15* | 22.02* | 38.07* | 64.24* |
| RE-GCN (SIGIR 2021) | 40.85* | 28.15* | 45.33* | 68.53* | 40.68* | 27.01* | 46.31* | 69.51* |
| DACHA (TKDD 2022) | 40.69* | 27.28* | 45.79* | 68.44* | 40.80* | 27.83* | 45.26* | 69.21* |
| TiRGN (IJCAI 2022) | 41.28* | 29.52* | 46.69* | 70.66* | 42.26* | 30.19* | 46.99* | 73.90* |
| EvoExplore (KBS 2022) | 28.11* | 13.97* | 33.45* | 57.67* | 29.82* | 18.50* | 30.08* | 58.01* |
| GTRL (TKDE 2024) | 38.57* | 27.36* | 42.15* | 66.35* | 38.43* | 27.48* | 43.06* | 67.82* |
| DHyper (TOIS 2024) | 41.71* | 29.37* | 45.69* | 69.32* | 42.84* | 29.96* | 47.52* | 70.82* |
| DECRL (NeurIPS 2024) | 42.90* | 30.49* | 47.06* | 70.01* | 43.36* | 30.64* | 47.96* | 71.12* |
| **HEDRA** | **47.86** | **35.28** | **53.32** | 75.65 | **46.77** | **33.66** | **52.33** | 75.64 |
| **Improvement** | **11.56%** | **15.71%** | **13.30%** | **2.16%** | **7.86%** | **9.86%** | **9.12%** | **-0.03%** |

## 5 EXPERIMENTS

### 5.1 EXPERIMENTAL SETUP

**Datasets and Experimental Settings.** HEDRA is evaluated on five widely adopted real-world TKG datasets: ICEWS14, ICEWS18, GDELT, WIKI, and YAGO (Trivedi et al., 2017; Li et al., 2021b). ICEWS14 and ICEWS18 are derived from the Integrated Crisis Early Warning System (Boschee et al., 2015), which records political events at daily granularity. GDELT, sourced from the Global Database of Events, Language, and Tone (Leetaru & Schrodt, 2013), captures human activities with timestamps at 15-minute intervals. WIKI and YAGO are organized at the yearly level, which are constructed from Wikipedia and YAGO3 (Mahdisoltani et al., 2013), respectively. The Mean Reciprocal Rank (MRR) and Hits@1/3/10 are adopted as evaluation metrics. Detailed statistics of the datasets and other experimental settings are provided in Appendix C.1. Unless otherwise stated, all loss-weight coefficients in HEDRA, including $\alpha_{\text{attn}}$ and $\beta_{\text{KL}}$, and $\lambda_{\text{align}}$ and $\lambda_{\text{sep}}$, are treated as fixed design choices and set to 0.5 in all experiments. They are introduced to balance the corresponding components and keep them on a comparable scale, rather than to serve as dataset-specific tuning knobs. A representative sensitivity study on $\alpha_{\text{attn}}$ and $\lambda_{\text{align}}$ on ICEWS14 dataset is reported in Appendix C.4, indicating that HEDRA is robust to moderate changes of these coefficients within a reasonable range.

**Baselines.** HEDRA is compared with eleven representative TKG representation learning approaches, grouped as follows: shallow encoder based approaches, i.e., TTransE (Leblay & Chekol, 2018) and TA-TransE (Garcia-Duran et al., 2018); GNN based approaches, i.e., RE-NET (Jin et al., 2020), Glean (Deng et al., 2020), RE-GCN (Li et al., 2021b), DACHA (Chen et al., 2021), and TiRGN (Li et al., 2022); and derived structure approaches, i.e., EvoExplore (Zhang et al., 2022), GTRL (Tang & Chen, 2024), DHyper (Tang et al., 2024), and DECRL (Chen & Chen, 2024). Detailed descriptions of these baselines are provided in Appendix C.2. Since some approaches do not target event prediction task and others collapse relations into four coarse-grained categories that differs from our settings, to ensure a fair comparison, we follow the official implementations of all baselines and tune hyperparameters to report the results.

### 5.2 PERFORMANCE COMPARISON

The performance of HEDRA with the compared approaches on the ICEWS14, ICEWS18, WIKI, YAGO, and GDELT datasets is presented in Tables 1, 2, and 3. It can be observed that HEDRA achieves average improvements of 5.70%, 7.51%, 7.21%, and 2.30% over the runner-up in terms of MRR and Hits@1/3/10 on five datasets, respectively. On ICEWS18 dataset, HEDRA yields a slightly lower Hits@10 than RE-NET. This difference can be attributed to RE-NET's global graph mechanism, which aggregates broader historical information and tends to retain more potentially relevant candidates within the top-10 range, whereas HEDRA is designed to emphasize event level causality disentanglement and improve the quality of the top ranks, leading to more pronounced gains in MRR, Hits@1, and Hits@3. Overall, these results show that disentangling heterogeneous

Table 2: The performance of HEDRA and the compared approaches on WIKI and YAGO. Since the YAGO dataset contains only 10 relation types, the Hits@10 metric is not statistically meaningful and is therefore denoted as "–". Other notations follow Table 1.

| Approach | WIKI | | | | YAGO | | | |
|---|---|---|---|---|---|---|---|---|
| | MRR | Hits@1 | Hits@3 | Hits@10 | MRR | Hits@1 | Hits@3 | Hits@10 |
| TTransE (WWW 2018) | 69.64* | 62.54* | 71.57* | 84.84* | 88.35* | 81.49* | 94.24* | – |
| TA-TransE (EMNLP 2018) | 70.76* | 66.26* | 74.32* | 88.47* | 87.16* | 85.53* | 90.50* | – |
| RE-NET (EMNLP 2020) | 75.29* | 57.08* | 90.25* | 97.64* | 92.24* | 89.43* | 92.43* | – |
| Glean (KDD 2020) | 91.76* | 86.18* | 90.55* | 92.84* | 90.19* | 88.90* | 91.71* | – |
| RE-GCN (SIGIR 2021) | 98.24* | 97.60* | 98.68* | 99.41* | 98.45* | 97.75* | 98.89* | – |
| DACHA (TKDD 2022) | 75.86* | 68.91* | 78.79* | 90.91* | 92.54* | 89.17* | 94.77* | – |
| TiRGN (IJCAI 2022) | 99.00* | 98.53* | 99.35* | 99.53* | 98.53* | 97.91* | 98.90* | - |
| EvoExplore (KBS 2022) | 78.71* | 73.13* | 81.42* | 88.43* | 93.92* | 91.47* | 95.21* | – |
| GTRL (TKDE 2024) | 92.68* | 89.18* | 92.34* | 95.63* | 92.36* | 90.71* | 93.95* | – |
| DHyper (TOIS 2024) | OOM | OOM | OOM | OOM | 94.38* | 92.03* | 96.01* | – |
| DECRL (NeurIPS 2024) | 93.20* | 90.91* | 94.33* | 98.14* | 95.84* | 94.15* | 97.09* | – |
| **HEDRA** | **99.14** | **98.73** | **99.48** | **99.79** | **99.12** | **98.77** | **99.31** | – |
| Improvement | 0.14% | 0.20% | 0.13% | 0.26% | 0.60% | 0.88% | 0.41% | – |

Table 3: The performance of HEDRA and the compared approaches on GDELT. "TLE" indicates a single epoch exceeded 24 hours. "OOM" indicates out of memory. Other notations follow Table 1.

| Approach | MRR | Hits@1 | Hits@3 | Hits@10 |
|---|---|---|---|---|
| TTransE (WWW 2018) | 15.09* | 4.71* | 13.69* | 39.72* |
| TA-TransE (EMNLP 2018) | 20.67* | 10.23* | 19.88* | 35.89* |
| RE-NET (EMNLP 2020) | TLE | TLE | TLE | TLE |
| Glean (KDD 2020) | 17.91* | 8.21* | 16.65* | 39.18* |
| RE-GCN (SIGIR 2021) | 21.35* | 11.20* | 21.73* | 44.53* |
| DACHA (TKDD 2022) | TLE | TLE | TLE | TLE |
| TiRGN (IJCAI 2022) | 22.46* | 12.10* | 22.33* | 45.89* |
| EvoExplore (KBS 2022) | 18.72* | 7.71* | 18.37* | 43.87* |
| GTRL (TKDE 2024) | 19.51* | 8.40* | 19.26* | 41.07* |
| DHyper (TOIS 2024) | OOM | OOM | OOM | OOM |
| DECRL (NeurIPS 2024) | 22.74* | 12.56* | 22.57* | 45.07* |
| **HEDRA** | **24.64** | **13.93** | **25.53** | **49.02** |
| Improvement | 8.36% | 10.91% | 13.11% | 6.82% |

event level causalities while discarding non-causality and spurious causality enables HEDRA to capture more discriminative entity and relation representations. The computational complexity of HEDRA can be found in Appendix C.3. The observed runtime increase is acceptable in light of the significant performance gains.

## 5.3 ABLATION STUDY

To assess the contribution of each component in HEDRA, ablation studies are performed on the ICEWS14 dataset, as shown in Table 4. Specifically, HEDRA-w/o-CDM removes the counterfactual detector module with the contrastive loss. HEDRA-w/o-EI and HEDRA-w/o-DD remove the event importance and the distributional discrep-

Table 4: The performance of HEDRA and its variants. The best performance is highlighted in **bold**.

| Approach | MRR | Hits@1 | Hits@3 | Hits@10 |
|---|---|---|---|---|
| HEDRA-w/o-CDM | 47.11 | 34.25 | 52.12 | 75.04 |
| HEDRA-w/o-EI | 47.34 | 34.65 | 52.33 | 75.39 |
| HEDRA-w/o-DD | 47.25 | 34.46 | 52.63 | 75.35 |
| HEDRA-w/o-IVDM | 46.47 | 33.77 | 51.65 | 74.75 |
| HEDRA-w/o-EOM | 46.24 | 33.49 | 51.79 | 74.10 |
| HEDRA | **47.86** | **35.28** | **53.32** | **75.65** |

ancy for constructing the non-causality mask, respectively. HEDRA-w/o-IVDM removes the IV-guided disentangling module with the robustness loss, and HEDRA-w/o-EOM excludes the evolutionary orthogonal module with the evolutionary loss. Since all ablated variants still share the same event level causality disentanglement framework, which provides a strong backbone compared with traditional entity level baselines, module-wise ablations tend to result in numerically modest performance drops and at the same time reflect the robustness of HEDRA.

The ablation results show that HEDRA-w/o-CDM exhibits only a modest performance drop, suggesting that the non-causality removed by the counterfactual detector module provides limited benefit to event prediction. In contrast, HEDRA-w/o-IVDM suffers a substantial performance degradation, demonstrating that the IV-guided disentangling module plays a critical role in eliminating spurious causality that impedes the model's acquisition of causally relevant discriminative informa-

Table 5: Top-5 predicted relations for two representative test samples on ICEWS14. Correctly predicted relations are indicated by a leading check mark (✓) and highlighted in **bold**.

| Test sample 1: ⟨Barack Obama, ?, Xi Jinping, 2014/11/13⟩ | |
| --- | --- |
| DECRL | HEDRA |
| ✓**Sign formal agreement** | ✓**Sign formal agreement** |
| Host a visit | Express intent to meet or negotiate |
| Express intent to meet or negotiate | ✓**Make statement** |
| ✓**Consult** | ✓**Make a visit** |
| ✓**Make statement** | ✓**Consult** |

| Test sample 2: ⟨Police (Hong Kong), ?, Protester (Hong Kong), 2014/11/29⟩ | |
| --- | --- |
| DECRL | HEDRA |
| ✓**Make statement** | ✓**Make statement** |
| Arrest, detain, or charge with legal action | Arrest, detain, or charge with legal action |
| Investigate | Fight with small arms and light weapons |
| Return, release person(s) | ✓**Use conventional military force** |
| ✓**Use conventional military force** | Investigate |

tion, which is essential for event prediction in TKGs. The hyperparameter sensitivity analysis can be found in Appendix C.4. The results indicate that the performance of HEDRA is insensitive to the history window length but shows a notable dependence on the neighbor count which is defined in the candidate graph.

## 5.4 CASE STUDY

Table 5 compares the predictions of HEDRA and the runner-up approach DECRL on two representative ICEWS14 test samples, one exhibiting a positive relational trend and the other a negative trend. Specifically, the table lists the top five predicted relations for test sample 1 ⟨Barack Obama, ?, Xi Jinping, 2014/11/13⟩ and test sample 2 ⟨Police (Hong Kong), ?, Protester (Hong Kong), 2014/11/29⟩. For test sample 1, more ground-truth relations are correctly identified by HEDRA, which predicts "Make a visit" rather than the inverse "Host a visit", indicating that event level causal disentangling better models relation directionality. By contrast, DECRL, which primarily captures entity correlations, struggles to distinguish relation directions. For test sample 2, only negative relations are predicted by HEDRA, whereas DECRL outputs the positive relation "Return, release person(s)", likely because it does not consider heterogeneous causalities at the event level in TKGs, which can lead to predictions with sentiment opposite to the ground-truth relations. In Appendix C.5, additional diagnostics of the training dynamics and IV-guided disentangling behavior are presented, together with few-shot robustness experiments on ICEWS14 and ICEWS18 datasets. Another case study can be found in Appendix C.6.

## 6 CONCLUSIONS AND FUTURE WORK

In this paper, based on a TKG structural causal model that establishes the theoretical framework for event level causality disentangling, a **H**eterogeneous **E**vent causality **D**isentangling **R**epresentation learning **A**pproach (**HEDRA**) is proposed, which is the first work that focuses on disentangling heterogeneous causalities at the event level in TKGs. Comprehensive experiments are conducted on five real-world datasets, including the comparison with baselines, ablation study, hyperparameter sensitivity analysis, running time analysis, training dynamics analysis, few-shot relations prediction, and case studies, which demonstrate the superior performance of HEDRA.

Future work includes replacing the fixed quantile rule in the IV-guided module with a learned, data-driven calibration mechanism for selecting genuine edges, designing lightweight global memory modules to enhance long-range history modeling and Hits@10 on large-scale datasets, and further reducing computational overhead via sparser causal subgraph construction, more efficient event causality disentangling, and sparsity-aware distributed implementations on datasets such as GDELT. Another promising direction is to combine HEDRA with LLM-based event prediction frameworks, using LLMs to provide semantic priors for event and relation representations and to leverage textual context for data-sparse and long-tail relations, while preserving explicit event level causality disentangling.

ACKNOWLEDGMENT

This work was funded by the National Key Research and Development Program of China (No. 2024YFB3312900).

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

# A    COMPREHENSIVE RELATED WORK

## A.1    TKG REPRESENTATION LEARNING

Current TKG representation learning approaches primarily aim to model correlations among entities or relations, which can be categorized into two main categories: modeling pairwise correlations between entities or relations and modeling high-order correlations among entities or relations.

TKG representation learning approaches modeling pairwise correlations typically combine Graph Convolutional Networks (GCNs) and Recurrent Neural Networks to capture structural information and temporal evolution, respectively (Li et al., 2021b; 2022; Bai et al., 2023). For example, RE-GCN (Li et al., 2021b) combines GCNs with Gated Recurrent Units (GRUs) to capture structural and temporal information, respectively. Similarly, TiRGN (Li et al., 2022) uses multi-relational GCNs and GRUs to capture the structural information of entities across different temporal patterns. These approaches generally stack multiple layers of networks to model pairwise correlations between distant entities, which can result in over-smoothing. To address this, some TKG representation learning approaches introduce paths or subgraphs to model pairwise correlations between distant entities or relations more effectively (Li et al., 2021a; Sun et al., 2021; Chen et al., 2024b; Zhang et al., 2024). For example, CluSTeR (Li et al., 2021a) and TITer (Sun et al., 2021) utilize reinforcement learning to discover cross-temporal clue paths that model pairwise correlations between entities. LogCL (Chen et al., 2024b) constructs both local and global subgraphs based on queries to achieve a similar goal.

In addition to pairwise correlations, high-order correlations among three or more entities or relations are modeled through derived structures, i.e., communities, hypergraphs, and evolutionary clusters (Zhang et al., 2022; Tang & Chen, 2024; Tang et al., 2024; Chen & Chen, 2024). For example, EvoExplore (Zhang et al., 2022) leverages dynamic communities based on soft modularity to model implicit correlations among multiple entities. DHyper (Tang et al., 2024) introduces hypergraph neural networks to capture high-order correlations among entities and among relations. DECRL (Chen & Chen, 2024) uses deep evolutionary clustering to construct evolutionary clusters for capturing high-order correlations among entities.

However, TKG datasets, e.g., ICEWS, centered on international political events, inherently involve complex causal dependencies. Focusing solely on entity or relation level correlations is therefore inadequate for event prediction in TKGs. To address this, we propose HEDRA, the first framework to disentangle heterogeneous causalities at the event level in TKGs.

## A.2    GRAPH CAUSALITY LEARNING

Graph causal learning approaches can be divided into static graph causal learning approaches and dynamic graph causal learning approaches.

Static graph causal learning approaches primarily model spatial causality in static graphs by generating interpretable subgraphs (Luo et al., 2020; Yuan et al., 2020; Ying et al., 2019; Fan et al., 2022; Gao et al., 2023). For example, PGExplainer (Luo et al., 2020) employs prior knowledge to pre-define subgraphs, thereby providing interpretability to static graphs. XGNN (Yuan et al., 2020) employs reinforcement learning to iteratively expand interpretable subgraphs. More recent approaches generate subgraphs using learnable methods. For example, GNNExplainer (Ying et al., 2019) uses mutual information to add or remove nodes and edges to generate subgraphs. RCGRL (Gao et al., 2023) relies on GNN-derived edge weights for generating subgraphs.

Table 6: The statistics of datasets.

| Dataset | #Entity | #Relation | #Training | #Validation | #Test | Interval |
|---------|---------|-----------|-----------|-------------|-------|----------|
| ICEWS14 | 7,128 | 230 | 74,845 | 8,514 | 7,371 | 24 hours |
| ICEWS18 | 23,033 | 256 | 373,018 | 45,995 | 49,545 | 24 hours |
| GDELT | 7,691 | 240 | 1,734,399 | 238,765 | 305,241 | 15 mins |
| WIKI | 12,554 | 24 | 539,286 | 67,538 | 63,110 | 1 year |
| YAGO | 10,623 | 10 | 161,540 | 19,523 | 20,026 | 1 year |

Dynamic graphs, prevalent in real-world scenarios, have motivated research on dynamic graph causal learning, which typically explores both spatial causality and temporal causality (Zhao & Zhang, 2024; Chen et al., 2024a). For example, DyGNNExplainer (Zhao & Zhang, 2024) utilizes disentangling methods to uncover spatio-temporal causality. CSI (Chen et al., 2024a) generates causal subgraphs by deriving query-related subgraphs and applying attention mechanisms to model spatio-temporal causality within dynamic graphs.

Despite progress in dynamic graph causal learning, most approaches mainly model static and dynamic causalities while overlooking spurious causality, which impedes the acquisition of causally relevant information for event prediction. Moreover, no theoretical framework exists to disentangle heterogeneous causalities at the event level in TKGs. To fill this gap, we propose HEDRA, which constructs event representations from quadruples and progressively disentangles non-causality, spurious causality, static causality, and dynamic causality among events in TKGs.

## B    BACKDOOR ADJUSTMENT DERIVATION

The detailed derivation process of backdoor adjustment is shown as follows:

$$
\begin{aligned}
P(\mathcal{Y}\,|\,\mathrm{do}(\mathcal{D})) &= \sum P\big(\mathcal{Y}\,|\,\mathrm{do}(\mathcal{D}),\mathcal{S}\big)\,P\big(\mathcal{S}\,|\,\mathrm{do}(\mathcal{D})\big) \\
&= \sum P\big(\mathcal{Y}\,|\,\mathrm{do}(\mathcal{C})\big)\,P(\mathcal{S}) \\
&= \sum P(\mathcal{S}) \sum P\big(\mathcal{Y}\,|\,\mathrm{do}(\mathcal{C}),\mathcal{T},\mathcal{P}\big)\,P\big(\mathcal{T},\mathcal{P}\,|\,\mathrm{do}(\mathcal{C})\big) \\
&= \sum P(\mathcal{S}) \sum P(\mathcal{Y}|\mathcal{G})\,P(\mathcal{T},\mathcal{P}) \\
&= \sum P(\mathcal{S}) \sum P(\mathcal{T}) \sum P(\mathcal{P}) \sum P(\mathcal{Y}|\mathcal{G}).
\end{aligned}
\tag{18}
$$

This derivation illustrates how the adjustment effectively blocks the backdoor paths involving $\mathcal{N}, \mathcal{P}$, and $\mathcal{S}$, thereby ensuring unbiased estimation of causal effects.

## C    EXPERIMENTS APPENDIX

### C.1    DATASET STATISTICS AND EXPERIMENTAL SETTINGS

Detailed statistics of the datasets are summarized in Table 6. HEDRA is implemented in Python with PyTorch and trained on an NVIDIA RTX 5090 GPU. The Neural Network Intelligence (NNI)[1] toolkit is employed to automatically explore hyperparameter configurations. The search spaces of key hyperparameters are defined as follows: $N_{\mathrm{layer}}$, the number of layers, ranges from 1 to 5 with a step size of 1; $N_{\mathrm{window}}$, the length of historical windows, ranges from 1 to 14 with a step size of 1; and $k$, the number of $k$-nearest neighbors in the candidate graph, ranges from 3 to 15 with a step size of 2. A maximum of 30 trials are conducted in the NNI search process, with the Tree-structured Parzen Estimator (Bergstra et al., 2015) employed as the optimization algorithm. The final selected hyperparameters are summarized in Table 7.

Hyperparameters $\lambda_{\mathrm{con}}, \lambda_{\mathrm{rob}}$, and $\lambda_{\mathrm{evo}}$ are all set to 0.1, controlling the magnitude of contrastive, robustness, and evolutionary losses, respectively. The Adam optimizer (Kingma & Ba, 2014) is applied with an initial learning rate of 0.01. The batch size is 16, and the representation dimension is 200. Results are averaged over five independent runs.

---

[1]https://github.com/microsoft/nni

Table 7: The final choices of key hyperparameter values.

| Hyperparameter | Search space | ICEWS14 | ICEWS18 | GDELT | WIKI | YAGO |
|---|---|---|---|---|---|---|
| $N_{\text{layer}}$ | $\{1, 2, 3, 4, 5\}$ | 2 | 2 | 1 | 1 | 2 |
| $N_{\text{window}}$ | $\{1, 2, 3, 4, 5, 6, 7, 8, 9, 10, 11, 12, 13, 14\}$ | 10 | 2 | 2 | 4 | 6 |
| $k$ | $\{3, 5, 7, 9, 11, 13, 15\}$ | 13 | 11 | 9 | 7 | 9 |

## C.2 DESCRIPTION OF BASELINES

To validate the effectiveness of HEDRA, we compare it against eleven representative TKG representation learning approaches, which are summarized as follows:

**Shallow Encoder based Approaches:**

- **TTransE** (Leblay & Chekol, 2018) augments TransE by embedding temporal information into entity representations.
- **TA-TransE** (Garcia-Duran et al., 2018) extends TransE by incorporating RNN-based modeling to capture time-aware relation representations.

**GNN based Approaches:**

- **RE-NET** (Jin et al., 2020) combines GCNs for capturing structural information together with RNNs to model temporal dependencies.
- **Glean** (Deng et al., 2020) leverages composition-based GCNs to encode entity interactions and employs GRUs to model temporal evolution.
- **RE-GCN** (Li et al., 2021b) integrates relation-aware GCNs with autoregressive GRUs to jointly capture structural and temporal dependencies.
- **DACHA** (Chen et al., 2021) introduces dual GCNs for structure information encoding and incorporates a self-attentive mechanism to learn relation-aware temporal representations.
- **TiRGN** (Li et al., 2022) introduces RGCNs to capture graph structural information and a double recurrent mechanism to model temporal dependencies.

**Structure Derived Approaches:**

- **EvoExplore** (Zhang et al., 2022) employs dynamic community structure to characterize the evolution of local structural patterns.
- **GTRL** (Tang & Chen, 2024) introduces group structure to model distant and indirectly connected entities, and integrates GRUs for temporal reasoning.
- **DHyper** (Tang et al., 2024) leverages hypergraph neural networks to model high-order dependencies among entities and relations.
- **DECRL** (Chen & Chen, 2024) represents the SOTA structure derived approach by employing deep evolutionary clustering to trace the temporal evolution of high-order correlations among entities.

## C.3 COMPLEXITY ANALYSIS

The time complexity of relation-aware GCN is $O((N_e + N_r)D^2)$, where $N_e$ and $N_r$ are the numbers of entities and relations, respectively. $D$ is the dimension of representations. The time complexity of the event representation construction module is $O(E^2D + ED^2)$, where $E$ is the number of events. The time complexity of the counterfactual detector module is $O(E^2D + ED^2)$. The time complexity of the IV-guided disentangling module is $O((E + N_r)D^2)$. The time complexity of the evolutionary orthogonal module is $O(ED^2)$. For the event prediction module, the time complexity is $O(D)$. Therefore, the total time complexity of HEDRA is $O(E^2D + (N_e + N_r + E)D^2)$.

A comparison of the per-epoch training and inference times of DHyper, DECRL, and HEDRA on the ICEWS14 dataset is provided in Table 8. Compared with DECRL, HEDRA improves MRR and Hits@1 by 11.56% and 15.71%, respectively, and provides more accurate predictions of relation directionality and sentiment, which are crucial for real-world applications. Plotted as latency–MRR

Table 8: Running time (in seconds) comparison.

| Approach | Traning time | Inference time |
|---|---|---|
| DHyper (TOIS 2024) | 636.85 | 91.43 |
| DECRL (NeurIPS 2024) | 907.47 | 142.82 |
| HEDRA | 1226.63 | 195.15 |

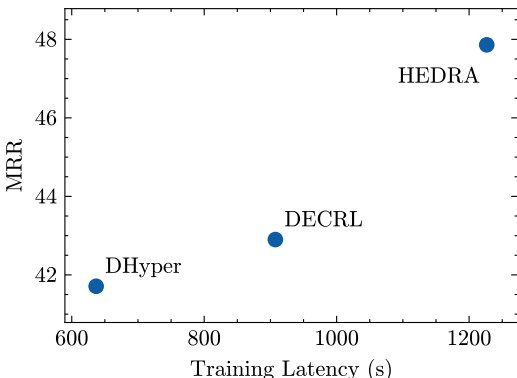

Figure 4: Latency–MRR Pareto frontier comparing DHyper, DECRL, and HEDRA.

points, these three approaches form a clear Pareto frontier, as shown in Figure 4. DHyper occupies the low-latency, lower-accuracy end, DECRL provides an intermediate trade-off, and HEDRA lies at the high-accuracy, higher-latency corner. On ICEWS14 dataset, HEDRA improves MRR over DHyper from $41.71$ to $47.86$, yielding an absolute gain of $6.15$ MRR points (approximately $15\%$ relative improvement), and over DECRL from $42.90$ to $47.86$, corresponding to an absolute gain of $4.96$ MRR points (approximately $12\%$ relative improvement). These gains come at the cost of roughly $2\times$ and $1.3\times$ higher training latency, respectively. Overall, this latency–accuracy trade-off suggests that the additional runtime of HEDRA is justified when higher predictive accuracy is prioritized over absolute latency.

To quantify the resource footprint of HEDRA across datasets, Table 9 summarizes the parameter counts, peak CUDA memory, and wall-clock training time on ICEWS14 and GDELT datasets. On ICEWS14 dataset, HEDRA uses 20.7M parameters, reaches a peak CUDA memory footprint of 2.95 GB, and requires 1226.63 s ($\approx$ 0.34 GPU hours) of training time, with an inference time of 195.15 s. On GDELT dataset, whose number of events is about $20\times$ that of ICEWS14 dataset, HEDRA uses 22.1M parameters, reaches a peak CUDA memory of 9.91 GB, and requires 14274.41 s ($\approx$ 3.97 GPU hours) of training time. Consequently, when moving from ICEWS14 to GDELT, the parameter count increases only modestly ($\sim$ $6\%$), peak memory grows by about $3.4\times$, and wall-clock training time increases by about $11.6\times$, which is substantially sub-linear in the dataset size increase.

## C.4 HYPERPARAMETER SENSITIVITY ANALYSIS

The sensitivity analysis of the key hyperparameters of HEDRA, i.e., the length of historical windows $N_{\text{window}}$ and the number of neighbors $k$, on the ICEWS14 dataset is shown in Figure 5. It can be observed that the length of historical windows has only a minor impact on performance, indicating that disentangling event level causalities enables HEDRA to learn robust entity and relation representations. In contrast, the number of neighbors has a significant effect, as it directly determines the size of the candidate graph. As $k$ increases, the performance gradually improves and then remains relatively stable when $k > 7$. In addition, Figure 6 reports a representative sensitivity study on the loss-weight coefficients $\alpha_{\text{attn}}$ and $\lambda_{\text{align}}$ on ICEWS14 dataset, using the same curve-based visualization, indicating that HEDRA is robust to moderate changes of these coefficients within a reasonable range and that the default symmetric setting around $0.5$ lies in a near-optimal region.

Table 9: Resource statistics of HEDRA on ICEWS14 and GDELT.

| Dataset | Prameters (M) | Peak CUDA memory (GB) | Trainning time (s) |
|---------|---------------|-----------------------|--------------------|
| ICEWS14 | 20.7 | 2.95 | 1226.63 |
| GDELT | 22.1 | 9.91 | 14274.41 |

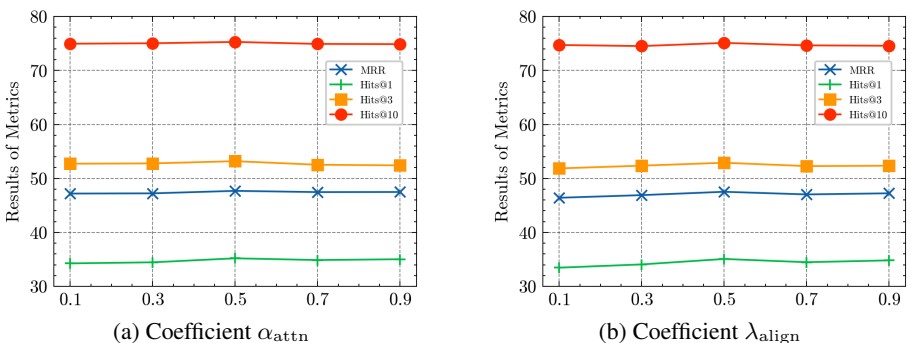

(a) Hyperparameter $N_{\text{window}}$       (b) Hyperparameter $k$

Figure 5: Hyperparameter sensitivity analysis.

## C.5 MODEL ANALYSIS

This subsection reports additional diagnostics on the training dynamics and IV-guided disentangling behaviour on ICEWS14 dataset, as well as the robustness of HEDRA on few-shot relations on ICEWS14 and ICEWS18 datasets.

**Training Dynamics and IV-guided Module.** Four loss terms are monitored during training: the event prediction loss $L_{\text{TKG}}$, the contrastive loss $L_{\text{con}}$, the robustness loss $L_{\text{rob}}$, and the evolutionary loss $L_{\text{evo}}$. To quantify how mass evolves on spurious causal edges inside the causal branch, a diagnostic statistic $\bar{p}_s$ is introduced as:

$$\bar{p}_s = \frac{1}{|\mathcal{E}_s|} \sum_{e \in \mathcal{E}_s} p_s(e), \tag{19}$$

where $\mathcal{E}_s$ denote the set of edges currently assigned to the spurious-causality set, and $p_s(e)$ denote the probability that edge $e$ belongs to spurious causality.

Table 10 reports the evolution of the loss terms and $\bar{p}_s$ over the 10 epochs on ICEWS14 dataset. All four losses decrease smoothly over epochs without noticeable oscillation or divergence, indicating that the interactions among the contrastive, robustness, and evolutionary losses keep the optimization process stable. $\bar{p}_s$ increases from 0.817 at epoch 1 to approximately 0.852 by epochs 7 and then saturates. Since the fraction of edges assigned to the spurious set is approximately fixed by construction, this increase in $\bar{p}_s$ indicates that the IV-guided module becomes progressively more

(a) Coefficient $\alpha_{\text{attn}}$       (b) Coefficient $\lambda_{\text{align}}$

Figure 6: Coefficient sensitivity analysis.

Table 10: Training dynamics and spurious-mass diagnostic on ICEWS14. The epoch marked with * corresponds to the best validation MRR.

| Epoch | $L_{\mathrm{TKG}}$ | $L_{\mathrm{con}}$ | $L_{\mathrm{rob}}$ | $L_{\mathrm{evo}}$ | $\bar{p}_s$ |
|---|---|---|---|---|---|
| 1 | 3.8341 | 0.5589 | 0.8724 | 0.3108 | 0.817 |
| 2 | 3.2442 | 0.2589 | 0.2644 | 0.0152 | 0.837 |
| 3 | 3.1236 | 0.2396 | 0.2350 | 0.0033 | 0.843 |
| 4 | 3.0461 | 0.2308 | 0.2262 | 0.0024 | 0.847 |
| 5 | 2.9763 | 0.2260 | 0.2210 | 0.0018 | 0.850 |
| 6 | 2.9166 | 0.2225 | 0.2182 | 0.0016 | 0.851 |
| 7 | 2.8641 | 0.2225 | 0.2159 | 0.0014 | 0.852 |
| 8* | 2.8230 | 0.2197 | 0.2150 | 0.0014 | 0.852 |
| 9 | 2.7895 | 0.2192 | 0.2140 | 0.0013 | 0.852 |
| 10 | 2.7730 | 0.2184 | 0.2137 | 0.0013 | 0.852 |

Table 11: Few-shot relation performance of HEDRA on ICEWS14 and ICEWS18.

| Dataset | MRR | Hits@1 | Hits@3 | Hits@10 |
|---|---|---|---|---|
| ICEWS14 | 26.68 | 11.96 | 29.79 | 62.63 |
| ICEWS18 | 17.44 | 8.53 | 14.64 | 38.44 |

confident about which edges are spurious and concentrates spurious-causality mass on them. As a result, these edges are more strongly down-weighted during message passing and decoding, which is the intended behaviour. The best validation MRR on ICEWS14 dataset is reached around epoch 8, when $\bar{p}_s$ has essentially flattened, suggesting that performance gains coincide with the model's improved ability to identify and suppress spurious causality, while later epochs mainly refine representations on top of this learned genuine-versus-spurious partition.

**Few-shot Relations.** To examine robustness on relations with limited supervision, a few-shot relation setting is constructed on ICEWS14 and ICEWS18 datasets. For each dataset, 20% of the relations are randomly selected, only 20% of their quadruples are retained for training, and evaluation is performed solely on this subset of relations. Under this highly sparse supervision, HEDRA achieves the results in Table 11, indicating that the model retains a non-trivial level of robustness for relations with limited training data.

### C.6 ANOTHER CASE STUDY

For each entity on the ICEWS14 dataset, all associated events are first grouped by timestamp, and only one event per timestamp is randomly chosen and retained. The most recent 20 distinct timestamps are then chronologically ordered to construct trajectories that characterize the temporal evolution of the entity. Figure 7 illustrates the stepwise magnitudes of change in both dynamic and static components between consecutive events. For China and Japan, the dynamic component is observed to vary more and by larger amounts than the static component, indicating that short-term shocks are absorbed by the dynamic component while long-term semantics remain stable in the static component, consistent with the goal of HEDRA to disentangle dynamic and static causalities.

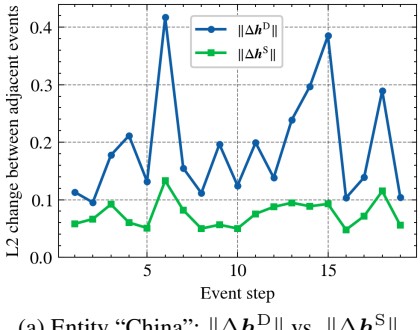 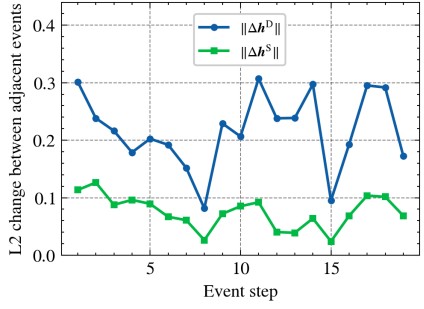

(a) Entity "China": $\|\Delta\boldsymbol{h}^{\mathrm{D}}\|$ vs. $\|\Delta\boldsymbol{h}^{\mathrm{S}}\|$.   (b) Entity "Japan": $\|\Delta\boldsymbol{h}^{\mathrm{D}}\|$ vs. $\|\Delta\boldsymbol{h}^{\mathrm{S}}\|$.

Figure 7: Case study of stepwise changes over the last 20 events. $\Delta$ denotes the first difference between consecutive events and $\|\cdot\|$ denotes the Euclidean norm.

