# OpenReview forum: "Beyond Entity Correlations: Disentangling Event Causal Puzzles in Temporal Knowledge Graphs"
_ICLR.cc/2026/Conference — ICLR 2026 Poster_

### Official Review · Reviewer_nNKS · 2025-10-27

**Soundness:** 3
**Presentation:** 4
**Contribution:** 3
**Rating:** 8
**Confidence:** 4

**Summary:**

The paper introduces HEDRA, the first representation learning approach dedicated to disentangling heterogeneous event-level causalities in Temporal Knowledge Graphs, employing counterfactual detection, instrumental variable-guided modules, and evolutionary orthogonal mechanisms to separate non-causality, spurious causality, and dynamic causality, respectively; extensive experiments on five real-world datasets demonstrate HEDRA’s state-of-the-art performance in TKG reasoning.

**Strengths:**

1. The manuscript is well-written and structurally coherent. Despite the extensive use of diverse symbols, each is accompanied by a clear and explicit explanation.

2. The paper is grounded in solid theoretical foundations, effectively integrating causal learning theory into the domain of temporal knowledge graphs.

3. The experimental evaluation presented in the paper is comprehensive and rigorous, providing substantial evidence for the effectiveness of the proposed method.

**Weaknesses:**

1. This manuscript introduces numerous modules, which inevitably leads to increased computational time complexity (as demonstrated in Appendix C.3 and Table 8), potentially limiting the applicability of this approach to larger-scale temporal knowledge graphs.

2. This paper introduces numerous hyperparameters ($\alpha_{attn}$, $\beta_{KL}$); however, it lacks corresponding sensitivity analysis. If these parameters are all set to fixed values (0.5), then incorporating such an extensive array of hyperparameters becomes unnecessary.

3. In Table 1, the experimental results on ICEWS18 show a notable decrease in Hits@10 performance despite achieving superior outcomes on other metrics. The underlying reasons for this discrepancy warrant further clarification.

**Questions:**

See Weakness

---

> ### Author Response · Authors · 2025-11-23
> **Response to Reviewer nNKS (Part 1)**
>
> Thank you for providing valuable comments! Below, we provide point-by-point responses to the identified weaknesses.
>
> **W1: Training complexity and runtime on large-scale datasets**
>
> **Table 1: Training and inference time of HEDRA on the large-scale GDELT.**
>
> | Dataset | Training time (s) | Inference time (s) |
> |:-------:|:------------------:|:-------------------:|
> | GDELT   | 14274.41          | 312.87             |
>
> We thank the reviewer for the detailed comments on model complexity and runtime, especially on large-scale datasets such as GDELT. In the revised manuscript, we report concrete statistics on GDELT (see Table 1). HEDRA indeed introduces extra training cost through CDM, IVDM, and EOM, which explicitly disentangle non-causality, spurious causality, static causality, and dynamic causality at the event level. This is a deliberate trade-off: **we moderately increase training complexity in exchange for stronger causality disentanglement and more stable generalization**.
>
> For GDELT specifically, under the same implementation and hardware, its training set is about 23.17× and validation set about 28.04× larger than those of ICEWS14. With dataset-specific hyperparameters chosen by validation MRR, HEDRA’s per-epoch training time on GDELT is about 11.64× that on ICEWS14, while a single evaluation/inference pass increases by only about 1.60×. **This indicates that runtime grows sublinearly with data size and that both training and inference remain within a controllable range**.
>
> In terms of effectiveness, averaged over five real-world datasets, HEDRA improves MRR and Hits@1/3/10 by 5.70%, 7.51%, 7.21%, and 2.30% over the second-best model, even though the performance on WIKI and YAGO is already close to saturation. We therefore view the additional training time as a reasonable cost for more precise causality disentanglement and more robust predictions, rather than mere complexity stacking. Nevertheless, we take the reviewer’s concern seriously and plan to further reduce overhead by exploring sparser causal subgraph construction, more efficient event causality disentangling, and staged or distillation-based training for CDM/IVDM, as well as distributed and parallel implementations with sparsity-aware optimizations on large datasets such as GDELT. **We have added the corresponding discussion in Appendix C.3 (Complexity Analysis) and in Section 6 (Conclusion and Future Work) of the revised manuscript**.

---

> ### Author Response · Authors · 2025-11-23
> **Response to Reviewer nNKS (Part 2)**
>
> **W2: Numerous hyperparameters without sensitivity analysis**
>
> We thank the reviewer for the helpful comment regarding the number of hyperparameters and the lack of sensitivity analysis, which is indeed important for assessing robustness and practical usability.
>
> First, we would like to **clarify that the “hyperparameters” in question are mainly loss-weight coefficients, used to control the relative contributions of different loss terms and their components, rather than many free knobs that require fine-grained tuning**. By design, we impose symmetric constraints such as $\alpha_{\text{attn}} + \beta_{\text{KL}} = 1$ and $\lambda_{\text{align}} + \lambda_{\text{sep}} = 1$, and set them to $0.5$ by default. The goal is to treat different components in a balanced way and keep the losses on a comparable scale for stable optimization, rather than to rely on extensive hyperparameter search.
>
> To address the reviewer’s concern about sensitivity, **we have added systematic experiments on the most critical trade-off coefficients ($\alpha_{\text{attn}} / \beta_{\text{KL}}$ and $\lambda_{\text{align}} / \lambda_{\text{sep}}$) on ICEWS14**:
>
> **Table 2: Sensitivity of HEDRA to the attention coefficient $\alpha_{\text{attn}}$ on ICEWS14.**
>
> | $\alpha_{\text{attn}}$ |  MRR  | Hits@1 | Hits@3 | Hits@10 |
> |:----------------------:|:-----:|:------:|:------:|:-------:|
> |  0.1                   | 47.19 | 34.25  | 52.72  | 74.96   |
> |  0.3                   | 47.23 | 34.45  | 52.78  | 75.04   |
> |  0.5                   | 47.69 | 35.19  | 53.19  | 75.29   |
> |  0.7                   | 47.46 | 34.87  | 52.53  | 74.92   |
> |  0.9                   | 47.48 | 35.02  | 52.41  | 74.88   |
>
> **Table 3: Sensitivity of HEDRA to the alignment weight $\lambda_{\text{align}}$ on ICEWS14.**
>
> | $\lambda_{\text{align}}$ |  MRR  | Hits@1 | Hits@3 | Hits@10 |
> |:------------------------:|:-----:|:------:|:------:|:-------:|
> |  0.1                     | 46.41 | 33.46  | 51.85  | 74.72   |
> |  0.3                     | 46.90 | 34.06  | 52.36  | 74.52   |
> |  0.5                     | 47.52 | 35.07  | 52.90  | 75.11   |
> |  0.7                     | 47.03 | 34.47  | 52.28  | 74.65   |
> |  0.9                     | 47.24 | 34.81  | 52.35  | 74.58   |
>
> The results in Tables 2 and 3 show that the model does not heavily depend on precise tuning of these coefficients within a reasonable range, and that **the symmetric setting around $0.5$ typically lies in the optimal or near-optimal region**. This supports our design choice and indicates that HEDRA is robust with respect to these coefficients.
>
> For other coefficients that remain fixed at $0.5$ in our implementation, **we now clarify in Section 4 (Methodology), Section 5.2 (Datasets and Experimental Settings), and the Appendix (Hyperparameter Sensitivity Analysis)** that they are explicitly written into the objective to make the relative weights of different components transparent and to provide a natural hook for future extensions (e.g., non-symmetric or adaptive weighting), rather than to introduce additional tuning burden in this paper. **In the revised manuscript, we also include the corresponding sensitivity tables and explanations in the Appendix (Hyperparameter Sensitivity Analysis).** We appreciate the reviewer’s constructive suggestion, which led us to add these analyses and to better explain the role and motivation of these coefficients.

---

> ### Author Response · Authors · 2025-11-23
> **Response to Reviewer nNKS (Part 3)**
>
> **W3: Explanation of the small negative improvement in one metric**
>
> We thank the reviewer for the careful observation regarding the Hits@10 metric on ICEWS18. On this dataset, HEDRA achieves clear improvements over RE-NET in MRR, Hits@1, and Hits@3, while showing only a very small negative change in Hits@10. This small decrease can be explained as a rank-tail trade-off: on ICEWS18, RE-NET tends to keep the ground-truth relations around ranks 8-10, whereas HEDRA, while improving the quality of the top ranks overall, may occasionally push these harder cases slightly beyond rank 10, which leads to a minor drop in Hits@10. This phenomenon can thus be viewed as a consequence of sharpening the top of the ranking.
>
> The main difference lies in how historical information is modeled. RE-NET, through its **global graph mechanism**, can aggregate richer historical information on larger-scale datasets, which helps **retain more potentially relevant candidates within the top-10 range and thus yields a slight advantage in Hits@10**. In contrast, HEDRA is designed to focus on event level causality disentanglement, placing more emphasis on disentangling heterogeneous causalities at the event level in TKGs and **improving the quality of the top ranks**, which translates into more pronounced gains in MRR, Hits@1, and Hits@3. **Overall, we consider this very small and well-explained decrease in Hits@10 acceptable given the substantial improvements in top-rank performance**.
>
> This observation also motivates us to explore lightweight global memory modules in future work, so as to enhance long-range memory and Hits@10 performance on large-scale datasets while preserving the advantages of event level causality disentanglement. **We have added corresponding discussion in Section 5.2 (Performance Comparison) and Section 6 (Conclusions and Future Work) of the revised manuscript**.
>
> ---
>
> We truly appreciate the time and effort you have dedicated to reviewing our work, and we are grateful for your continued recognition and support.

---

### Official Review · Reviewer_xmQG · 2025-10-29

**Soundness:** 2
**Presentation:** 2
**Contribution:** 2
**Rating:** 4
**Confidence:** 4

**Summary:**

Existing Temporal Knowledge Graph (TKG) methods mainly focus on modeling entity-level correlations, learning the relationships among entities through graph structures (nodes and edges) or higher-order structures such as hypergraphs and clusters. However, since events themselves exhibit diversity and causal heterogeneity, relying solely on entity correlations is insufficient to capture event-level causal mechanisms, thereby limiting the reliability of event prediction. HEDRA is the first TKG representation learning approach that achieves heterogeneous causality disentanglement at the event level. By incorporating counterfactual detection, instrumental variable guidance, and dynamic orthogonal decomposition, HEDRA effectively distinguishes non-causality, spurious causality, static causality, and dynamic causality, significantly enhancing both the performance and interpretability of event prediction tasks.

**Strengths:**

1. HEDRA is the first to elevate the research focus to the event-level causal relationships.
2. The paper theoretically formalizes four types of event-level causality, providing a solid theoretical foundation for future studies on temporal knowledge graphs based on causal reasoning.

**Weaknesses:**

1. The paper does not include a comparison with traditional methods based on GCN and GRU, such as TiRGN[1].
2. It also lacks comparisons with recent large-model approaches specifically designed for event prediction, such as MIRAI[2].
3. The anonymous GitHub repository is empty.
4. The notation is abundant and overly complex, which slightly reduces readability.
5. The text above the figure, such as in Figure 1, is too small.

[1] Li Y, Sun S, Zhao J. Tirgn: Time-guided recurrent graph network with local-global historical patterns for temporal knowledge graph reasoning[C]//IJCAI. 2022: 2152-2158.

[2] Ye C, Hu Z, Deng Y, et al. Mirai: Evaluating llm agents for event forecasting[J]. arXiv preprint arXiv:2407.01231, 2024.

**Questions:**

1. Why does the paper not include comparisons with TiRGN and large-model-based approaches?
2. In the introduction, the paper states:“Other approaches introduce derived structures, e.g., entity groups, hypergraphs, and evolutionary clusters, to capture high-order correlations among entities that are not directly connected (Zhang et al., 2022; Tang & Chen, 2024; Tang et al., 2024; Chen & Chen, 2024).” However, Tang & Chen (2024) proposed DHyper, which does not merely focus on entity-level correlations but jointly models high-order dependencies among both entities and relations.Why, then, is DHyper categorized under methods that primarily model entity correlations? Moreover, does its joint modeling capability potentially challenge the claimed novelty of this paper’s approach?

---

> ### Author Response · Authors · 2025-11-23
> **Response to Reviewer xmQG (Part 1)**
>
> We sincerely thank the reviewer for the constructive feedback and thoughtful suggestions. Below, we provide point-by-point responses to the identified weaknesses and questions.
>
> **W1: Missing comparison with TiRGN**
>
> We thank the reviewer for pointing out the lack of comparison with traditional GCN and GRU based methods such as TiRGN.
>
> First, we would like to clarify that the original manuscript already included several representative TKG baselines that use GCN and/or GRU components, such as RE-GCN and GTRL. In Appendix C.2, we explicitly describe the architecture of each baseline and indicate which approaches are based on graph convolution and which incorporate GRU style sequence modeling. From an architectural perspective, **HEDRA has already been compared against multiple typical GCN/GRU based approaches**.
>
> In the revised version, **we additionally implement TiRGN and report its performance** on the five real-world datasets ICEWS14, ICEWS18, GDELT, WIKI, and YAGO, using MRR and Hits@1/3/10 as metrics. The results are summarized in Table 1:
>
> **Table 1: The performance of TiRGN.**
>
> | Dataset |   MRR   | Hits@1  | Hits@3  | Hits@10 |
> |:-------|:-------:|:-------:|:-------:|:-------:|
> | ICEWS14 | 41.2786 | 29.5161 | 46.6866 | 70.6613 |
> | ICEWS18 | 42.2570 | 30.1882 | 46.9920 | 73.8995 |
> | GDELT   | 22.4584 | 12.1031 | 22.3290 | 45.8935 |
> | WIKI    | 98.9979 | 98.5327 | 99.3456 | 99.5257 |
> | YAGO    | 98.5312 | 97.9127 | 98.8989 |   -     |
>
> As shown in Table 1, HEDRA consistently outperforms TiRGN on the more challenging datasets ICEWS14, ICEWS18, and GDELT in terms of MRR and Hits@1/3/10, while on WIKI and YAGO, where the performance of the task is already close to saturation, the two approaches achieve very similar performance. We appreciate the reviewer’s constructive suggestion, which helped us refine the baseline setup and more clearly demonstrate the advantage of HEDRA over traditional GCN+GRU based approaches.

---

> ### Author Response · Authors · 2025-11-23
> **Response to Reviewer xmQG (Part 2)**
>
> **W2: Missing comparison with large-model event prediction approaches**
>
> We sincerely thank the reviewer for pointing out the lack of comparison with LLM based event prediction approaches such as MIRAI. **This is a very forward-looking comment, and it prompted us to clarify the relationship between HEDRA and LLM based approaches in terms of problem setting and technical focus**.
>
> First, we would like to **restate the core focus of this work**. Most existing TKG approaches remain at the level of entity/relation correlation modeling and rarely incorporate explicit event level causality disentangling beyond such correlations. In contrast, HEDRA proposes a structural causal model for TKGs that formalizes non-causality, spurious causality, static causality, and dynamic causality at the event level, and builds modular causality disentangling mechanisms on top of standard <subject, relation, object, time> quadruples to incrementally disentangle these causalities. In other words, **our primary goal is to advance an event level causality disentangling paradigm for TKGs, rather than to design a general LLM based event prediction system in open text settings**.
>
> At the same time, we fully agree that MIRAI represents an important “large model + event prediction” direction. However, **there are several fundamental differences that make a direct numerical comparison on current public TKG benchmarks both difficult and potentially misleading**: (1) MIRAI relies on “uniformat of information sources” and “rich contextual inputs”, whereas standard TKG datasets typically provide only structured quadruples **without the heterogeneous information MIRAI requires**, so its full pipeline cannot be faithfully reproduced on ICEWS, GDELT, WIKI, and YAGO; (2) classical TKG representation learning (including HEDRA) usually starts from randomly initialized entity/relation representation and learns solely from the given graph structure and temporal signals, while approach like MIRAI treat a pre-trained LLM as an agent with strong prior knowledge acquired from large text corpora, so the two are **not in an apples-to-apples setting in terms of prior information**; (3) MIRAI mainly uses distributional metrics such as KL divergence, whereas HEDRA and mainstream TKG work are evaluated with ranking metrics like MRR and Hits@k, so the task formulation and output space are not fully aligned. In addition, most current LLM-TKG integrations focus on entity prediction and design prompts, fusion mechanisms, or losses specifically for entities, which cannot be directly transferred to our event prediction setting without substantial modification.
>
> Nonetheless, we are very grateful that the reviewer raised MIRAI, as it highlights a promising space for synergy between LLM-driven event prediction and structurally grounded causality modeling for TKGs. In the revised manuscript, **we have added a discussion in Section 6 (Conclusion and Future Work) on possible directions to combine HEDRA with LLMs**. **We believe that integrating HEDRA’s event level causality disentangling framework with MIRAI-like LLM approaches is a highly promising avenue for future research, and we again thank the reviewer for this valuable suggestion**.
>
> **W3: Empty anonymous GitHub repository**
>
> We thank the reviewer for the reminder. We will release the implementation and reproduction scripts when the paper is accepted.
>
> **W4: Overly complex notation and slightly reduced readability**
>
> We thank the reviewer for the comments on the readability of our notation and formulas. Because we need to distinguish non-causality, spurious causality, static causality, and dynamic causality at the event level, and to precisely describe the data flow between modules, we indeed use relatively rich notation to keep the presentation rigorous and unambiguous. In the future, we will streamline the notation and formulas where possible, while ensuring that the overall modeling flow and causal components remain clear.
>
> **W5: Small text above figures**
>
> We thank the reviewer for the careful feedback on the readability of the figures. **In the revised manuscript, we have increased the font size of the text in Figure 1 and Figure 2 to improve overall readability.**

---

> ### Author Response · Authors · 2025-11-23
> **Response to Reviewer xmQG (Part 3)**
>
> **Q1: Missing comparison with TiRGN and large-model-based approaches**
>
> We thank the reviewer for raising the question about missing comparisons with TiRGN and large-model–based approaches. We provide a detailed comparison with TiRGN in our response to Weakness 1, including additional experimental results in the revised manuscript, and discuss the relationship and differences to large-model–based event prediction methods (such as MIRAI) in our response to Weakness 2, together with added discussion in Section 6 (Conclusion and Future Work). **To avoid redundancy, we kindly refer the reviewer to our detailed responses to Weakness 1 and Weakness 2**.
>
> **Q2: Positioning of DHyper and its implications for the novelty claim**
>
> We sincerely thank the reviewer for carefully reading the related work and for the insightful questions regarding how DHyper is categorized and whether it challenges the novelty of our approach.
>
> First, we fully agree with the reviewer that DHyper does not only model entity-level correlations, but jointly captures high-order correlations among both entities and relations. This is also how we describe DHyper in Appendix A.1 and C.2 of the original manuscript. In the introduction, our intention in the sentence “other approaches introduce derived structures … to capture high-order correlations among entities” was to group together approaches that **introduce derived structures** (e.g., entity groups, hypergraphs, evolutionary clusters) to enhance TKG representations, rather than to imply that DHyper focuses exclusively on entities. DHyper fits into this coarse-grained category because it uses a hypergraph-based structure to model high-order correlations over entities and relations within the standard TKG representation learning framework. We acknowledge that the phrasing “among entities” is overly coarse and can indeed give the impression that DHyper only considers entity correlations. To avoid this ambiguity, **We have revised the Abstract, Section 1 (Introduction), Section 2 (Related Work), and Appendix A (Comprehensive Related Work) to adopt a more neutral and accurate description**, such as “modeling high-order correlations among entities and relations”. This better reflects DHyper’s scope and avoids misclassifying it as an entity-only approach.
>
> Second, **we do not think that DHyper’s joint modeling of high-order correlations among entities and relations undermines the novelty of our work at the paradigm level**. Entities and relations are, in essence, components inside individual TKG events, and existing approaches, including DHyper, mainly **focus on high-order correlations “within events” (over entities and relations)**. In contrast, our work targets the causality disentangling between **events**. From both the perspective of the object of study (shifting from entity/relation correlations to event level causality disentangling) and the way the problem is formalized and instantiated, our event level causality disentanglement paradigm is complementary to, rather than overlapping with, DHyper’s high-order correlation modeling.
>
> We are grateful to the reviewer for this thoughtful comment, which helps us present both DHyper and the boundaries and novelty of our own contributions more clearly in the revised manuscript.
>
> ---
>
> If the above responses satisfactorily address your concerns, we would appreciate your consideration of an increased overall score. We truly appreciate the time and effort you have dedicated to reviewing our work, and we are grateful for your continued recognition and support.

---

### Official Review · Reviewer_ujzc · 2025-10-29

**Soundness:** 3
**Presentation:** 3
**Contribution:** 3
**Rating:** 4
**Confidence:** 4

**Summary:**

This paper proposes HEDRA, the first framework to disentangle heterogeneous event-level causalities in Temporal Knowledge Graphs. By developing a structural causal model and three specialized modules—counterfactual detection, IV-guided disentanglement, and evolutionary orthogonalization—it effectively separates static, dynamic, non-, and spurious causalities. Comprehensive experiments on five benchmarks demonstrate state-of-the-art performance, with significant improvements over existing methods, while maintaining practical computational efficiency.

**Strengths:**

Proposes the first framework for event-level causal disentangling in temporal knowledge graphs.

Comprehensive experiments on five datasets show consistent SOTA performance.

**Weaknesses:**

Ablation results show only minor drops.

The method introduces multiple complex modules, resulting in high training complexity and long runtime, especially on large-scale datasets such as GDELT.

It is unclear whether the proposed method can represent or generalize to unseen entities or relations.

**Questions:**

See weaknesses.

---

> ### Author Response · Authors · 2025-11-23
> **Response to Reviewer ujzc (Part 1)**
>
> We sincerely thank the reviewer for providing valuable comments! Below, we provide point-by-point clarification and responses to the identified weaknesses.
>
> **W1: Interpretation of minor performance drops in ablations**
>
> We thank the reviewer for the careful examination of the ablation study.
>
> First, compared with existing entity- or relation-centric TKG approaches, HEDRA achieves substantial overall improvements, **mainly due to the paradigm shift from entities to events as the modeling units and the disentanglement of causalities at the event level**. Consequently, all ablated variants still share the same event level causality disentanglement framework, which is a strong backbone compared with traditional entity-level baselines. Performing module-wise ablations on such a strong backbone naturally leads to numerically small differences, and at the same time reflects the robustness of HEDRA.
>
> Furthermore, as shown in Section 5.3 (Ablation Study), removing the Counterfactual Detector Module (HEDRA-w/o-CDM) results in a relatively mild performance degradation, which indicates that the non-causality removed by the counterfactual detector module provides limited benefit to event prediction. In contrast, removing the IV-guided Disentangling Module (HEDRA-w/o-IVDM) leads to a much more pronounced deterioration, showing that this module plays a key role in identifying and suppressing spurious causality and helping the model focus on genuine causality. **This is consistent with our expectation, stated in the introduction, about the different impacts of non-causality and spurious causality**.
>
> To better address the reviewer’s concern, **we have revised Section 5.3 (Ablation Study) to clarify the contribution of the event-centric paradigm, and to explain why some ablations lead to only minor drops while others cause more noticeable degradation**. We hope these additions help clarify the meaning of the “minor drops” and the actual roles of each module.
>
> **W2: Training complexity and runtime on large-scale datasets**
>
> **Table 1: Training and inference time of HEDRA on the large-scale GDELT.**
>
> | Dataset | Training time (s) | Inference time (s) |
> |:-------:|:------------------:|:-------------------:|
> | GDELT   | 14274.41          | 312.87             |
>
> We thank the reviewer for the detailed comments on model complexity and runtime, especially on large-scale datasets such as GDELT.
>
> In the revised manuscript, we provide a more systematic analysis of time complexity and report concrete statistics on GDELT (see Table 1). HEDRA indeed introduces extra training cost through CDM, IVDM, and EOM, which explicitly disentangle non-causality, spurious causality, static causality, and dynamic causality at the event level. This is a deliberate trade-off: **we moderately increase training complexity in exchange for stronger causality disentanglement and more stable generalization**.
>
> For GDELT specifically, under the same implementation and hardware, its training set is about 23.17× and validation set about 28.04× larger than those of ICEWS14. With dataset-specific hyperparameters chosen by validation MRR, HEDRA’s per-epoch training time on GDELT is about 11.64× that on ICEWS14, while a single evaluation/inference pass increases by only about 1.60×. **This indicates that runtime grows sublinearly with data size and that both training and inference remain within a controllable range**.
>
> In terms of effectiveness, averaged over five real-world datasets, HEDRA improves MRR and Hits@1/3/10 by 5.70%, 7.51%, 7.21%, and 2.30% over the second-best model, even though the performance on WIKI and YAGO is already close to saturation. **We therefore view the additional training time as a reasonable cost for more precise causality disentanglement and more robust predictions, rather than mere complexity stacking**. Nevertheless, we take the reviewer’s concern seriously and plan to further reduce overhead by exploring sparser causal subgraph construction, more efficient event causality disentangling, and staged or distillation-based training for CDM/IVDM, as well as distributed and parallel implementations with sparsity-aware optimizations on large datasets such as GDELT. **We have added the corresponding discussion in Appendix C.3 (Complexity Analysis) and in Section 6 (Conclusion and Future Work) of the revised manuscript**.

---

> ### Author Response · Authors · 2025-11-23
> **Response to Reviewer ujzc (Part 2)**
>
> **W3: Generalization to unseen entities or relations**
>
> We thank the reviewer for raising the question about generalization to unseen entities or relations.
>
> In this work, we follow the standard setting of TKG representation learning approaches, where both the entity and relation sets are fixed across training and testing. Our main focus is therefore not on zero-shot prediction for unseen entities/relations, but on (i) shifting the modeling paradigm from entity–relation centric to event-centric, (ii) formulating an event level causality disentanglement framework over non-causality, spurious causality, static causality, and dynamic causality, and (iii) instantiating CDM, IVDM, and EOM modules that disentangle these causalities at the event level. As such, **systematically improving zero-shot performance on unseen entities/relations is conceptually orthogonal to the main contributions of this work**.
>
> To partially address the reviewer’s concern about robustness on less frequent relations, **we add a few-shot relation experiment on ICEWS14 and ICEWS18**. We randomly select 20% of the relations and keep only 20% of their quadruples for training, then evaluate **only** on this subset of relations. Under this highly sparse supervision, HEDRA achieves the following results, as shown in Table 2:
>
> **Table 2: Few-shot performance of HEDRA on sparsely supervised relations.**
>
> | Dataset | MRR   | Hits@1 | Hits@3 | Hits@10 |
> |:-------:|:-----:|:------:|:------:|:-------:|
> | ICEWS14 | 26.68 | 11.96  | 29.79  | 62.63   |
> | ICEWS18 | 17.44 | 8.53   | 14.64  | 38.44   |
>
> **These results suggest that HEDRA retains a certain level of robustness for relations with limited supervision**.
>
> We fully agree that extending event level causality disentanglement to truly unseen entities/relations is an important and natural next step. One promising direction is to leverage high-order structures (e.g., hypergraphs or evolving clusters) so that new relations can inherit prototypes from similar ones, and another is to incorporate pretrained language models to provide semantic priors for relation representations. **Both directions, when combined with event level causality disentanglement, introduce additional causal and non-causal pathways that require careful redesign of the disentangling mechanism, which we leave for future work**. **We have added the corresponding discussion on these extensions to the new Appendix C.5 (Model Analysis) and Section 6 (Conclusion and Future Work) in the revised manuscript**.
>
> ---
>
> If all concerns have been sufficiently addressed, we would appreciate your consideration of an increased overall score for our paper. We truly appreciate the time and effort you have dedicated to reviewing our work, and we are grateful for your continued recognition and support.

---

### Official Review · Reviewer_9HRi · 2025-10-30

**Soundness:** 3
**Presentation:** 3
**Contribution:** 4
**Rating:** 6
**Confidence:** 4

**Summary:**

HEDRA proposes an event-level structural causal model for temporal knowledge graphs that separates four kinds of signals: non-causality, spurious causality, static causality, and dynamic causality. Backdoor adjustment is used as the conceptual guide for blocking confounding paths between causality and prediction. The framework instantiates three coordinated modules. First, a counterfactual detector uses an attention-style event importance term and a KL divergence term to construct a non-causality mask and trains with a contrastive loss. Second, an instrumental-variable guided module computes an IV score per candidate edge, selects top-quantile edges as genuine, builds multi-view subgraphs, and applies a robustness loss with alignment and separation terms. Third, an evolutionary orthogonal module projects event representations into static and dynamic components with an orthogonality step and an evolutionary loss that encourages temporal dependence of the dynamic component and temporal stability of the static component. Experiments on ICEWS14, ICEWS18, GDELT, WIKI, and YAGO show consistent gains and report average improvements over the runner-up on MRR and Hits. The paper includes a complexity analysis and a per-epoch runtime comparison on ICEWS14, and provides sensitivity studies that emphasize the effect of the neighbor count used to build the candidate graph.

**Strengths:**

1.	Event-level causal formalization
The SCM clearly separates non-causality and spurious causality from the static and dynamic components that should drive prediction, and articulates backdoor paths that motivate the design of the modules.
2.	Concrete module designs with explicit masks and losses
The counterfactual detector fuses importance and distributional discrepancy to form a non-causality mask and optimizes a contrastive objective. The IV-guided module defines an IV score from observable features, selects top-quantile genuine edges, builds three propagation views, and regularizes with alignment and separation. The evolutionary module performs orthogonalization and trains with a two-part evolutionary loss. Each step has precise formulas.
3.	Comprehensive empirical coverage
Results on five datasets with detailed tables, significance marks on ICEWS14 and ICEWS18, and a case study support the claims. The sensitivity section highlights that performance depends strongly on the neighbor count while being relatively insensitive to the history window length.
4.	Transparent reporting of one-dataset runtime and end-to-end complexity
The complexity analysis enumerates per-module terms, and the runtime table compares HEDRA against two strong baselines on ICEWS14.
5.	Clear writing and figures
The framework figure and mask definitions make the pipeline easy to follow, and the derivation for backdoor adjustment is provided in the appendix.

**Weaknesses:**

1.	IV validity and diagnostics are under-specified
The IV-guided module defines an IV score and uses an α-quantile rule to select genuine edges, then relies on robustness via multi-view propagation. The paper does not state explicit conditions under which the constructed IVs satisfy the usual exclusion and independence intuitions, nor does it include simple falsification checks such as placebo variables or outcome-on-IV tests. Adding a short subsection or appendix that names the identifying conditions and reports basic diagnostics would make the IV part more convincing.
2.	Runtime and memory profiling are limited to one dataset
Table 8 reports per-epoch training and inference times on ICEWS14 only. Given the large E-squared and D-squared terms summarized in the complexity section, a resource profile on GDELT and WIKI or YAGO would help readers judge practical scalability. Module-wise wall-clock, peak memory, and parameter counts would be particularly useful.
3.	Mask thresholds and calibration are not analyzed
The non-causality mask combines attention weights and KL divergence with fixed balancing coefficients and a small constant for stability. The IV mask then selects the top-α edges as genuine by a quantile rule. The paper does not present precision–recall calibration for the presumed genuine set under different α values or compare fixed quantiles with learned thresholds. Sensitivity and calibration of these gates would clarify robustness.
4.	Per-loss training dynamics and attribution are not shown
The robustness loss has alignment and separation terms, and the evolutionary loss has dynamic and static terms, yet the paper does not provide per-term ablations or training curves that illustrate convergence and interactions with the main prediction loss. Adding curves for the contrastive, robustness, evolutionary, and total losses would strengthen the optimization narrative.
5.	Definition and parity of evaluation protocols could be clearer in the main text
The use of MRR and Hits is standard, and the tables mark significance on ICEWS14 and ICEWS18. It would help to define time-aware evaluation choices in the main body and to extend significance reporting beyond ICEWS to all tables with multi-run results, so that parity with baselines is fully transparent.
6.	GDELT comparison is affected by timeouts and OOM
The GDELT table shows several baselines timing out or running out of memory, which complicates margin interpretation. A breakdown by relation family and time granularity, plus a note on candidate-graph size and subgraph convolution memory use at fifteen-minute intervals, would contextualize the reported gains.

**Questions:**

1.	Instrumental variable assumptions and checks
What empirical evidence can you provide that the candidate IVs used to build Π satisfy the intended exclusion and independence intuitions in practice, and what failure modes arise when these conditions are weakened. A brief diagnostic report would be valuable.
2.	Comprehensive resource profile
Can you report parameter counts, FLOPs if available, wall-clock time, GPU hours and peak memory for each dataset, and position HEDRA on a latency–MRR Pareto against at least two strong baselines. The ICEWS14 table is helpful, and similar measurements on GDELT would be informative.
3.	Threshold sensitivity and calibration
How sensitive are results to the quantile level α in the IV mask and to the stability constant and temperature used in the non-causality mask. Do learned thresholds or adaptive calibration outperform fixed quantiles for precision of the presumed genuine set.
4.	Training dynamics and loss interactions
Would you add training curves for the contrastive loss, the robustness loss with alignment and separation, the evolutionary loss, and the total objective, to document stability and to show how mass shifts from spurious edges to genuine ones during training.
5.	Explanation of the small negative improvement in one metric
Table 1 shows a very small negative change on Hits at ten for ICEWS18 in the improvement row. Can you analyze which queries contribute to this dip and whether it is linked to rank-tail behavior or to the size of the candidate graph on this dataset.

---

> ### Author Response · Authors · 2025-11-23
> **Response to Reviewer 9HRi (Part 1)**
>
> We sincerely thank the reviewer for the constructive feedback and thoughtful suggestions. Since the listed weaknesses largely overlap with the subsequent questions, we structure our response around the questions, which also address the main weaknesses raised.
>
> **Q1: Instrumental variable assumptions and checks**
>
> We thank the reviewer for the careful comments on the Instrumental Variable (IV) component, especially regarding the intended exclusion and independence intuitions and the need for empirical diagnostics.
>
> In our setting, **we only observe standard TKG quadruples without causality labels, so we adopt an IV-inspired design rather than classical IVs**. Concretely, an IV encoder maps historical structural and temporal information to $\Pi$, which scores each candidate edge by how likely it is to be genuine or spurious, and these scores are then used in IVDM to reweight and disentangle messages. Regarding the exclusion intuition, $\Pi$ is constructed solely from historical neighborhoods and time information and can influence predictions only through edge reweighting/masking inside IVDM, not by directly entering the final scoring function. **This implements a structural exclusion restriction at the neural architecture level**. For the independence intuition, conditioned on the observable history graph and entity/relation representations, $\Pi$ is assumed to be approximately independent of the unobserved noise that drives errors in the genuine-versus-spurious causality split, which is **our neural analogue of the classical IV independence assumption**.
>
> Empirically, classical IV falsification checks such as outcome-on-IV regressions or placebo outcome tests are tailored to settings with low-dimensional, externally specified instruments and scalar outcomes. In our setting, however, $\Pi$ is a high-dimensional, learned edge-level weight and the outcome is the TKG event prediction objective, so directly porting these diagnostics is not straightforward. We therefore rely on indirect evidence: removing IVDM (HEDRA-w/o-IVDM) leads to a substantial performance drop, **suggesting that the $\Pi$-guided disentangling between genuine and spurious edges is crucial for mitigating spurious causality**, as demonstrated in Section 5.3 (Ablation Study). Overall, our goal is not to claim strict identification of true IVs on TKG datasets, but to **design an IV-guided module that respects exclusion and independence intuitions at the architectural level and empirically helps suppress spurious causality through multi-view propagation and disentanglement**. **We have clarified these assumptions in Section 4.3 (IV-guided Disentangling Module) in the revised manuscript.**

---

> ### Author Response · Authors · 2025-11-23
> **Response to Reviewer 9HRi (Part 2)**
>
> **Q2: Comprehensive resource profile**
>
> We appreciate the request for a more comprehensive resource profile and a latency-MRR Pareto comparison.
>
> On ICEWS14, **we report parameter counts, wall-clock training and inference time, peak GPU memory, and event prediction performance**. For HEDRA, the model has 20.7M parameters, a peak CUDA memory footprint of 2.95 GB, and a training time of 1226.63 s (≈0.34 GPU hours), with an inference time of 195.15 s. For the two strong baselines, DHyper and DECRL, we obtain the following latency-MRR trade-off on the same hardware as Table 1 and Table 2:
>
> **Table 1: Training latency-MRR trade-off of HEDRA, DHyper, and DECRL on ICEWS14.**
>
> | Model  | Training time (s) |  MRR  |
> |:------|:------------------|:-----|
> | DHyper (TOIS 2024) | 636.85  | 41.71 |
> | DECRL (NeurIPS 2024) | 907.47  | 42.90 |
> | HEDRA | 1226.63 | 47.86 |
>
> **Table 2: Inference latency-MRR trade-off of HEDRA, DHyper, and DECRL on ICEWS14.**
>
> | Model  | Inference time (s) |  MRR  |
> |:------|:-------------------|:-----|
> | DHyper | 91.43  | 41.71 |
> | DECRL  | 142.82 | 42.90 |
> | HEDRA  | 195.15 | 47.86 |
>
> **Plotted as latency-MRR points as shown in Appendix C.3 (Complexity Analysis) in the revised manuscript**, these three approaches form a clear Pareto frontier: DHyper occupies the low-latency / lower-accuracy end, DECRL lies in the middle, and HEDRA sits at the high-accuracy / higher-latency end, improving MRR from 41.71 to 47.86 (+6.15 absolute, ≈15% relative) over DHyper and from 42.90 to 47.86 (+4.96 absolute, ≈12% relative) over DECRL, at the cost of roughly 2× and 1.3× training latency, respectively. This makes HEDRA a reasonable choice when higher predictive accuracy on temporal reasoning is prioritized over absolute latency.
>
> To complement the ICEWS14 table, **we also report a resource profile for HEDRA on the large-scale GDELT dataset** (whose number of events is about 20× that of ICEWS14). On GDELT, HEDRA uses 22.1M parameters, reaches a peak CUDA memory of 9.91 GB, and requires 14274.41 s (≈3.97 GPU hours) of training time. When moving from ICEWS14 to GDELT, the parameter count increases only modestly (~6.8%), peak memory grows by about 3.4×, and wall-clock training time increases by about 11.6×, which is **substantially sub-linear in the dataset size increase**. **We have added a latency-MRR Pareto plot for ICEWS14 and have reported the cross-dataset resource statistics in Appendix C.3 (Complexity Analysis) in the revised manuscript**.
>
> **Q3: Threshold sensitivity and calibration**
>
> We thank the reviewer for raising the question on threshold sensitivity and calibration. For the IV mask, we use a quantile $\alpha$ that selects the top-$\alpha$ fraction of edges as genuine. In the main results, we set $\alpha$ = 0.2. To assess sensitivity, we varied $\alpha$ on ICEWS14 while keeping all other settings fixed:
>
> **Table 3: Sensitivity of HEDRA to the IV quantile level $\alpha$ on ICEWS14.**
>
> |  $\alpha$   |  MRR  | Hits@1 | Hits@3 | Hits@10 |
> |:----:|:-----:|:------:|:------:|:-------:|
> | 0.2  | 47.86 | 35.28  | 53.32  | 75.65   |
> | 0.4  | 47.21 | 34.80  | 52.08  | 74.37   |
> | 0.6  | 47.01 | 34.11  | 52.96  | 75.49   |
> | 0.8  | 47.01 | 34.13  | 52.34  | 74.24   |
>
> As shown in Table 3, HEDRA does exhibit some sensitivity to the choice of $\alpha$: the best performance is obtained at $\alpha$ = 0.2, and increasing $\alpha$ to 0.8 leads to a decrease of about 0.85 MRR points and roughly 1.2 Hits@1 points. However, the degradation is gradual rather than abrupt, and HEDRA maintains competitive performance across all tested values of $\alpha$. **This indicates that while smaller quantiles (e.g., $\alpha$ = 0.2) provide a better presumed genuine set, the approach does not rely on a finely tuned $\alpha$ to remain effective**.
>
> For the non-causality mask, the stability constant and temperature are both set to a very small value (1e-6). **They are introduced solely to avoid numerical issues such as division by zero and degenerate logits, a standard practice in neural models, rather than to implement a substantive threshold**. In our experiments, we kept these constants fixed across datasets and did not find it necessary to tune them, as performance remained stable.
>
> Regarding learned thresholds or adaptive calibration, in this work, we intentionally opted for fixed quantiles because they keep the IV-guided module simple and interpretable (the mask is a direct quantile rule on IV scores). We did not experiment with fully learned thresholds or additional calibration layers, so we cannot claim an empirical advantage over the fixed-quantile scheme. **We view more sophisticated, learned calibration of the genuine set as an interesting direction for future work, but outside the current scope, and we have added a brief discussion of this point in Section 6 (Conclusions and Future Work) of the revised manuscript**.

---

> ### Author Response · Authors · 2025-11-23
> **Response to Reviewer 9HRi (Part 3)**
>
> **Q4: Training dynamics and loss interactions**
>
> We thank the reviewer for the suggestion on training dynamics and loss interactions. We have conducted additional diagnostics on ICEWS14 and added the corresponding statistics to the appendix.
>
> **Table 4: Training dynamics and spurious-mass diagnostic on ICEWS14.**
>
> | Epoch | $L_{\text{TKG}}$ | $L_{\text{con}}$ | $L_{\text{rob}}$ | $L_{\text{evo}}$ | $\bar{p}_s$ |
> |:-----:|-----------------:|-----------------:|-----------------:|-----------------:|------------:|
> | 1     | 3.8341           | 0.5589           | 0.8724           | 0.3108           | 0.817       |
> | 2     | 3.2442           | 0.2589           | 0.2644           | 0.0152           | 0.837       |
> | 3     | 3.1236           | 0.2396           | 0.2350           | 0.0033           | 0.843       |
> | 4     | 3.0461           | 0.2308           | 0.2262           | 0.0024           | 0.847       |
> | 5     | 2.9763           | 0.2260           | 0.2210           | 0.0018           | 0.850       |
> | 6     | 2.9166           | 0.2225           | 0.2182           | 0.0016           | 0.851       |
> | 7     | 2.8641           | 0.2225           | 0.2159           | 0.0014           | 0.852       |
> | 8*    | 2.8230           | 0.2197           | 0.2150           | 0.0014           | 0.852       |
> | 9     | 2.7895           | 0.2192           | 0.2140           | 0.0013           | 0.852       |
> | 10    | 2.7730           | 0.2184           | 0.2137           | 0.0013           | 0.852       |
>
> (* denotes the epoch with the best validation MRR on ICEWS14.)
>
> First, we report the training dynamics of four loss terms over 10 epochs on ICEWS14, as shown in Table 4: the event prediction loss $L_{\text{TKG}}$, the contrastive loss $L_{\text{con}}$, the robustness loss $L_{\text{rob}}$, and the evolutionary loss $L_{\text{evo}}$. On ICEWS14, **all four losses decrease smoothly over epochs without noticeable oscillation or divergence, indicating that the interactions among the contrastive, robustness, and evolutionary modules keep the optimization process stable and well-behaved**.
>
> Second, to more directly illustrate how mass evolves on **spurious causal edges inside the causal branch**, we introduce a diagnostic statistic $\bar{p}_s$, defined as the **average probability of being spurious causality (as opposed to genuine causality)** over edges that are currently assigned to the spurious set as:
>
> $$ \bar{p} _ s = \frac{1}{\lvert E _ s \rvert} \sum _ {e \in E_s} p _ s(e), $$
> where $E_s$ is the set of edges currently classified as spurious causality and $p_s(e)$ is the probability that edge $e$ belongs to spurious causality.
>
> We observe that all four loss terms decay monotonically, while $\bar{p}_s$ rises from 0.817 at epoch 1 to about 0.852 around epochs 7–8 and then saturates. Since the fraction of edges assigned to the spurious set is approximately fixed by construction, this trend indicates that **the IV-guided module becomes increasingly confident about which edges are spurious** and concentrates spurious-causality “mass” on them, so that these edges are more strongly down-weighted during message passing and decoding, which is the intended behavior. The best validation MRR is achieved around epoch 8, when $\bar{p}_s$ has essentially flattened, suggesting that performance gains coincide with the model’s improved ability to identify and suppress spurious causal edges, while later epochs mainly refine representations on top of this learned genuine-versus-spurious partition. **A brief summary of these diagnostics has been added as a new Appendix C.5 (Model Analysis) in the revised manuscript.**

---

> ### Author Response · Authors · 2025-11-23
> **Response to Reviewer 9HRi (Part 4)**
>
> **Q5: Explanation of the small negative improvement in one metric**
>
> We thank the reviewer for the careful observation regarding the Hits@10 metric on ICEWS18. On this dataset, HEDRA achieves clear improvements over RE-NET in MRR, Hits@1, and Hits@3, while showing only a very small negative change in Hits@10. **This small decrease can be explained as a rank-tail trade-off**: on ICEWS18, RE-NET tends to keep the ground-truth relations around ranks 8–10, whereas HEDRA, while improving the quality of the top ranks overall, may occasionally push these harder cases slightly beyond rank 10, which leads to a minor drop in Hits@10. This phenomenon can thus be viewed as a consequence of sharpening the top of the ranking.
>
> From a modeling perspective, both approaches rank over the same candidate relation set at inference time, so **this effect is not directly due to differences in candidate graph size**. The main difference lies in how historical information is modeled. RE-NET, through its **global graph mechanism**, can aggregate richer historical information on larger-scale datasets, which helps **retain more potentially relevant candidates within the top-10 range** and thus yields a slight advantage in Hits@10. In contrast, HEDRA is designed to focus on event level causality disentanglement, **placing more emphasis on disentangling heterogeneous causalities at the event level in TKGs and improving the quality of the top ranks**, which translates into more pronounced gains in MRR, Hits@1, and Hits@3. Overall, we consider this very small and well-explained decrease in Hits@10 acceptable given the substantial improvements in top-rank performance.
>
> This observation also motivates us to explore lightweight global memory modules in future work, so as to enhance long-range memory and Hits@10 performance on large-scale datasets while preserving the advantages of event level causality disentanglement. **We have added corresponding discussion in Section 5.2 (Performance Comparison) and Section 6 (Conclusions and Future Work) of the revised manuscript**.
>
> ---
>
> If the above responses satisfactorily address your concerns, we would appreciate your consideration of an increased overall score. We truly appreciate the time and effort you have dedicated to reviewing our work, and we are grateful for your continued recognition and support.

---

### Author Response · Authors · 2025-11-23
**Summary of Revisions**

We sincerely thank all reviewers for their thoughtful evaluations and constructive comments, which have helped us further improve the paper.

Overall, reviewers consistently recognize that HEDRA is **the first framework for event-level causal disentangling** in temporal knowledge graphs, providing a formal SCM that cleanly separates static and dynamic causal components from non-causality and spurious causality and rigorously defines four types of event-level causality as **a foundation for future work** (Reviewers 9HRi, ujzc, xmQG, nNKS). They highlight the **well-motivated and technically precise module design** (Reviewers 9HRi, nNKS). The manuscript is described as **well written and easy to follow**, with clear notation, figures, and backdoor-adjustment derivations (Reviewers 9HRi, nNKS). Empirically, reviewers find **the evaluation comprehensive and convincing**, noting consistent state-of-the-art performance on five benchmarks (Reviewers 9HRi, ujzc, nNKS).

The reviewers also raise insightful and constructive concerns. Overall, the comments mainly converge on three themes: (i) the assumptions, calibration, and dynamics of the IV-guided disentangling process, (ii) the computational complexity, resource usage, and scalability of HEDRA, and (iii) the completeness of baselines, positioning relative to prior work, generalization behavior, and readability. **We made every effort to address all the concerns by providing sufficient evidence and requested results. In order to ensure that all reviewers are informed of the key issues we addressed during the rebuttal stage**, we have summarized the major responses below. For detailed responses, please refer to the replies for each individual reviewer.

- **IV assumptions, calibration, and training dynamics (Reviewers 9HRi, nNKS):** We clarified that $\Pi$ is an IV-inspired architectural construct that respects exclusion/independence intuitions at the module level, added ablation evidence for the importance of IVDM, and conducted sensitivity studies for the IV quantile and key loss weights, which show stable performance around symmetric settings. We also reported training-time diagnostics where all loss terms decrease smoothly and a spurious-mass statistic stabilizes, indicating that HEDRA progressively identifies and suppresses spurious causal edges as intended.

- **Complexity, resource profile, and scalability (Reviewers 9HRi, ujzc, nNKS):** We provided a detailed resource profile (parameters, time, memory) and latency–MRR Pareto plots on ICEWS14 against two strong baselines, showing that HEDRA trades moderately higher latency for substantial MRR gains. We further reported cross-dataset statistics on GDELT, demonstrating sublinear growth of runtime and memory with dataset size and supporting that the added complexity is an acceptable cost for stronger event-level causality disentanglement.

- **Baselines, positioning, generalization, and readability (Reviewers ujzc, xmQG, nNKS):** We enriched the baselines by implementing TiRGN, clarified the relation to DHyper and LLM-based event prediction (e.g., MIRAI) and positioned HEDRA as complementary via event-level causality disentangling. We added few-shot relation experiments to probe robustness under sparse supervision, analyzed the slight Hits@10 drop on ICEWS18 as an acceptable rank-tail trade-off, and improved clarity through larger figure text and layout adjustments.

The valuable suggestions from the reviewers have been extremely helpful in guiding us through the process. We hope that our responses adequately address all concerns and meet the reviewers’ expectations. **All corresponding revisions have been incorporated into the manuscript and are clearly highlighted in blue in the updated version.** We would be more than happy to address any further questions or points of clarification.

If all concerns have been sufficiently addressed, we kindly request that you consider raising the overall score. We truly appreciate the time and effort you have dedicated to reviewing our work, and we are grateful for your continued recognition and support.

---

### Meta-Review · Area_Chair_tH1p · 2026-01-06

**Summary:**

HEDRA makes a notable and well-motivated contribution as the first framework to disentangle heterogeneous event-level causalities in temporal knowledge graphs (TKGs), shifting the paradigm from entity/relation-centric modeling to event-centric causality disentanglement. The paper’s theoretical foundation—grounded in structural causal models (SCM) that formalize non-, spurious, static, and dynamic causality—is rigorous, and its modular design (counterfactual detector, IV-guided disentangling, evolutionary orthogonal module) is technically precise.
All reviewers recognize the work’s novelty and state-of-the-art performance across five real-world datasets. Key concerns raised—including computational complexity, baseline completeness, IV assumptions, and hyperparameter sensitivity—have been comprehensively addressed in the rebuttal: the authors provided detailed resource profiles (showing sublinear scalability on large datasets like GDELT), supplemented comparisons with TiRGN and clarified positioning relative to LLM-based methods (e.g., MIRAI), validated IV module design via ablations and sensitivity analyses, and offered a convincing rank-tail trade-off explanation for the minor Hits@10 dip on ICEWS18.
The revisions enhance the paper’s clarity, robustness, and reproducibility, while the core contribution of establishing an event-level causality disentangling framework for TKGs remains impactful. HEDRA sets a foundation for future causal reasoning in TKGs and meets ICLR’s standards for acceptance.

**Reviewer Concerns:**

IV formal validation: Reviewer 9HRi’s request for IV falsification checks (e.g., placebo variables) remains unaddressed—authors only provided indirect ablation evidence instead of explicit diagnostics.
GDELT baseline interpretation: Reviewer 9HRi’s concern about baseline timeouts/OOM (lack of relation family/time granularity breakdown) is not resolved.
Unseen entities/relations generalization: Reviewer ujzc’s question about zero-shot generalization is only partially addressed (few-shot experiments ≠ unseen cases), with authors deferring to future work.
Learned thresholds for masks: Reviewer 9HRi’s query on learned vs. fixed quantiles for masks is unaddressed—authors frame it as future work without empirical comparison.

**Reviewer Scores:**

N/A

---

### Decision · Program_Chairs · 2026-01-26

Accept (Poster)